# Plasticity of gene expression in the nervous system by exposure to environmental odorants that inhibit HDACs

**Sachiko Haga-Yamanaka[1†], Rogelio Nunez-Flores[1,2†], Christi A Scott[3†], Sarah Perry[4], Stephanie Turner Chen[3], Crystal Pontrello[1], Meera G Nair[2], Anandasankar Ray[1,3,4]\***

[1]Department of Molecular, Cell and Systems Biology, University of California, Riverside, United States; [2]Division of Biomedical Sciences, University of California, Riverside, United States; [3]Cell, Molecular and Developmental Biology Program, University of California, Riverside, United States; [4]Genetics, Genomics and Bioinformatics Program, University of California, Riverside, United States

**Abstract** Eukaryotes respond to secreted metabolites from the microbiome. However, little is known about the effects of exposure to volatiles emitted by microbes or in the environment that we are exposed to over longer durations. Using *Drosophila melanogaster,* we evaluated a yeast-emitted volatile, diacetyl, found at high levels around fermenting fruits where they spend long periods of time. Exposure to the diacetyl molecules in headspace alters gene expression in the antenna. *In vitro* experiments demonstrated that diacetyl and structurally related volatiles inhibited conserved histone deacetylases (HDACs), increased histone-H3K9 acetylation in human cells, and caused changes in gene expression in both *Drosophila* and mice. Diacetyl crosses the blood–brain barrier and exposure caused modulation of gene expression in the mouse brain, therefore showing potential as a neuro-therapeutic. Using two separate disease models previously known to be responsive to HDAC inhibitors, we evaluated the physiological effects of volatile exposure. Diacetyl exposure halted proliferation of a neuroblastoma cell line in culture. Exposure to diacetyl vapors slowed progression of neurodegeneration in a *Drosophila* model for Huntington's disease. These changes strongly suggest that certain volatiles in the surroundings can have profound effects on histone acetylation, gene expression, and physiology in animals.

## eLife assessment

This interesting and **important** work shows that diacetyl, a volatile organic compound released by yeast in fermenting fruit, can act as a histone deacetylase (HDAC) inhibitor and trigger wide changes in gene expression, together with suppression neurotoxicity in a *Drosophila* model of Huntington's disease. While the effects on gene expression changes and degenerative phenotypes are **convincingly** shown, further studies are required to determine whether and how olfactory sensory neurons and odorant receptors mediate the effects of diacetyl described by the authors.

## Introduction

Multicellular organisms evolve in the presence of microbial volatiles and metabolites (**McFall-Ngai et al., 2013**; **Dorrestein et al., 2014**), often exposed for prolonged periods of time. Human-associated

**\*For correspondence:** anand.ray@ucr.edu

[†]These authors contributed equally to this work

**Preprint posted** 21 February 2023
**Sent for Review** 21 February 2023
**Reviewed preprint posted** 02 May 2023
**Reviewed preprint revised** 02 February 2024
**Version of Record published** 27 February 2024

microbes have incredible metabolic diversity, encoding 100× as many genes as the human genome itself (**McFall-Ngai et al., 2013**). It is estimated that about half of the thousands of metabolites found in mammalian blood are either produced or altered by microbes (**Wikoff et al., 2009**). Microbial metabolites are an important mechanism by which the microbiome influences host immunity amongst other things (**Dorrestein et al., 2014**; **Rooks and Garrett, 2016**). Apart from microbial volatiles, many volatiles are also generated from the metabolism of a variety of food components, like triglycerides, sugars, and amino acids (**Shibamoto, 2014**). Some chemicals are also produced by microorganisms such as yeasts and lactic bacteria during fermentation in many foods and beverages (**Shibamoto, 2014**).

In the model system *Drosophila melanogaster*, prolonged exposure to $CO_2$ showed an increase in volume of the $CO_2$-sensing glomerulus in the antennal lobe, which lead to adaptations in fly behavior (**Devaud et al., 2001**; **Sachse et al., 2007**). Carbon dioxide is a common by-product of yeast fermentation in overripe fruits, the preferred food source. In order to study the physiological effects of long-term exposure to other volatiles emitted during fermentation, we exposed flies for 5 d to diacetyl, a buttery odorant made at high levels by yeast in fermenting fruits (1350 ppb in ripe banana), that is also found in many commonly consumed foods such as butter, yoghurt, wine, fruits, beer, and popcorn (**Shibamoto, 2014**; **Martineau et al., 1995**; **Nykanen and Nykanen, 1991**; **Hughes and Baxter, 2001**). Diacetyl was selected for two reasons: first, *D. melanogaster* use fermenting fruits as their favored food source, oviposition site, and spend long durations of time exposed to it. Second, diacetyl is known to be an inhibitory odorant of the $CO_2$ receptor and therefore an opposite effect is expected on the V-glomerulus from the previous study (**Turner and Ray, 2009**). Follow-up experiments revealed that diacetyl acts as a histone deacetylase (HDAC) inhibitor *in vitro*. We found that volatiles structurally like diacetyl have similar HDAC inhibitory properties as well. Subsequent analyses with two different model systems – *D. melanogaster* and *Mus musculus* – showed gene expression changes in tissues caused by odor exposure from a distant source. We found expression of hundreds of genes to be modulated upon exposure to the volatile, as would be expected from an inhibitor of HDACs. HDACs are histone-modifying enzymes involved in the removal of acetyl groups from lysine residues and the remodeling of chromatin structure to regulate gene expression (**Shahbazian and Grunstein, 2007**; **Gräff and Tsai, 2013**). With large-scale roles in gene regulation, HDACs are promising targets in drug development for many diseases such as cancers and neurodegenerative disorders (**Minucci and Pelicci, 2006**; **Kazantsev and Thompson, 2008**; **Chuang et al., 2009**). Several classes of orally administered HDAC inhibitors have been found to attenuate the progression of certain cancers and neurodegenerative diseases, including Alzheimer's disease and Huntington's disease (**Chuang et al., 2009**). We determined that exposure to diacetyl volatiles substantially altered gene expression in the mouse brain, presumably by crossing the blood–brain barrier. Diacetyl exposure also slowed degeneration of photoreceptor cells in a Huntington's disease model in *Drosophila*. Last, exposure to diacetyl caused dramatic reduction in proliferation of a neuroblastoma cell line in culture. Our discovery of a family of volatile odorants that also act as HDAC inhibitors highlights the need to understand how small molecules present in our environment interact with and alter eukaryotic gene expression.

## Results
### Activity-dependent expression modulation of chemosensory receptors
Previous studies demonstrated that prolonged exposure of ~5 d to elevated $CO_2$ levels causes an increase in volume of the $CO_2$-sensitive V-glomerulus in the antennal lobe of *Drosophila*, which lead to adaptations in fly behavior (**Devaud et al., 2001**; **Sachse et al., 2007**). To assess the effect of the $CO_2$ response inhibitor diacetyl on the antennal lobe, flies were exposed to the odorant during the first 5 days immediately following eclosion. The $CO_2$-detecting ab1C neuron was labeled using the promoters of the $CO_2$ receptors, *Gr63a-Gal4; UAS-mcd8GFP* and *Gr21a-Gal4; UAS-mcd8GFP*, and the flies were exposed to headspace above a 1% (V/V) diacetyl solution in an air-tight container. The flies were then imaged for expression of GFP in the ab1C neurons of the antenna, as well as the V glomerulus, which receives axonal input from ab1C neurons in the antennal lobe (**Jones et al., 2007**). Surprisingly, the *Gr63a* promoter-driven GFP signal in the V glomerulus was completely abolished after 5 days of exposure to the odor for all brains imaged (**Figure 1A**). However, there was also a concomitant decrease in GFP signal in ab1C neurons in the antenna (**Figure 1A and B**). Similar results

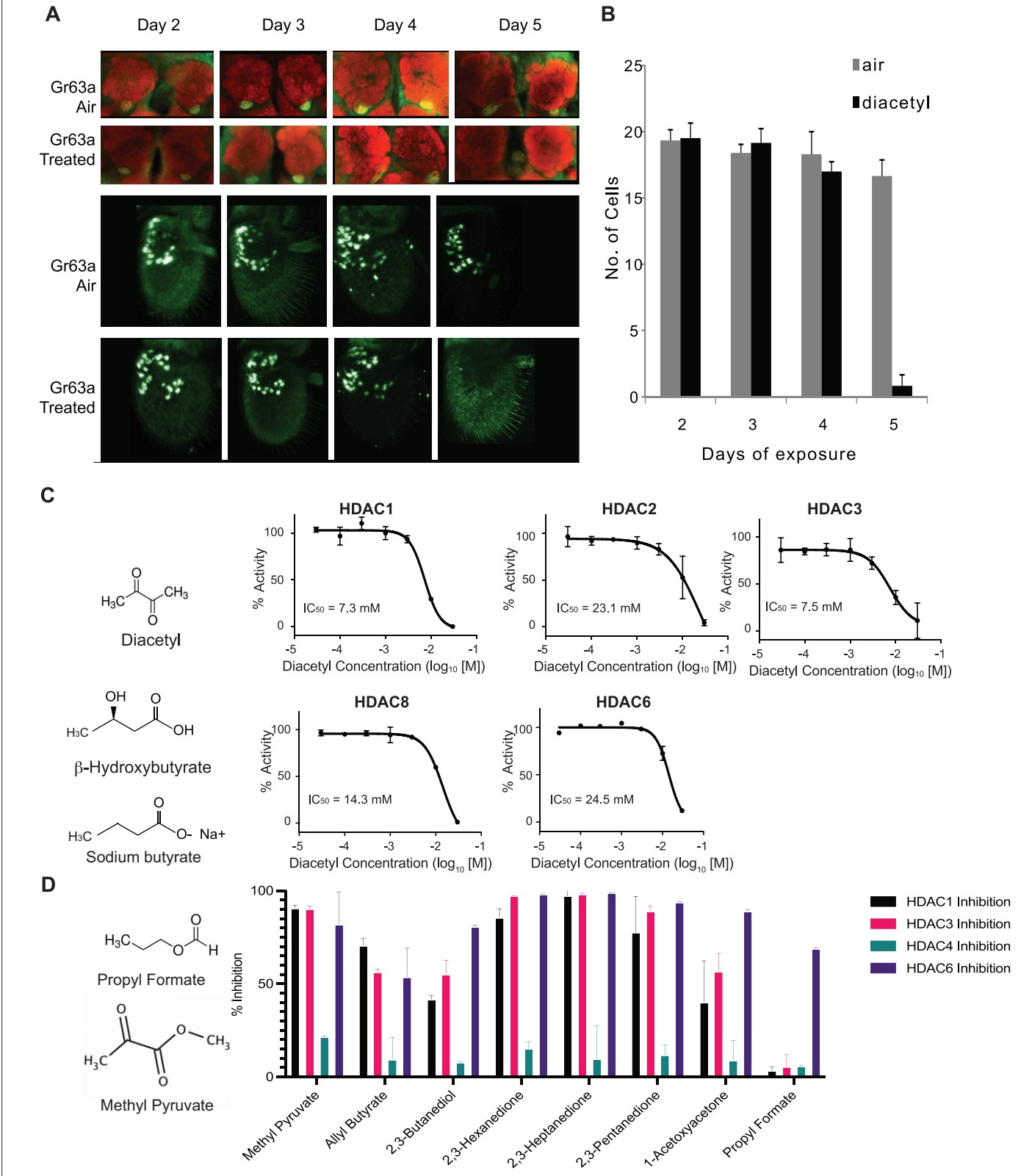

**Figure 1.** Volatiles found in microbes can inhibit histone deacetylases (HDACs) *in vitro*. (**A**) Antennal and whole-mount brain staining of *Gr63a-Gal4* flies. Flies were exposed to diacetyl (headspace from $10^{-2}$ soln) or air for 2–5 d. Brains and antenna were dissected on the indicated days, fixed, and then stained for neuropil marker nc82 (red) and anti-GFP (green). Magnification 25 x. (**B**) Mean of ab1C neurons expressing GFP after indicated days of odor exposure. d4on = 2,3-butanedione. n = 6, error bars = SEM. Schematic chemical structures of diacetyl, β-hydroxybutyrate, and sodium butyrate.

*Figure 1 continued on next page*

*Figure 1 continued*

(**C**) Dose–activity curves of class I HDACs: HDAC1, HDAC2, HDAC3, HDAC8, and class II HDAC6 treated with various concentrations of diacetyl. Percentage of HDAC activity is relative to the activity of each enzyme without diacetyl. IC50s are indicated in the chart areas. Error bars = SEM, n = 4–5.

(**D**) Representative structures of odorants that inhibit HDACs (left), and average percentage inhibition of class I HDACs: HDAC1, HDAC3, and class II HDAC4, HDAC6 treated with 15 mM of indicated volatiles (right). Error bar = SD, each tested in duplicate.

The online version of this article includes the following figure supplement(s) for figure 1:

**Figure supplement 1.** $CO_2$ inhibitory odor diacetyl causes downregulation of $CO_2$ receptor.

**Figure supplement 2.** Downregulation of the $CO_2$ receptor driver is reversible.

were seen for *Gr21a*-driven expression, where expression of GFP was lost in both the V-glomerulus and ab1C neurons (***Figure 1—figure supplement 1A and B***). Interestingly, after 5 d of recovery in clean air, the GFP expression was regained in the V-glomerulus as well as in ab1C neurons in the antenna, indicating that both *Gr21a* and *Gr63a* expression had been restored (***Figure 1—figure supplement 2***). Because olfactory neurons do not regenerate in the fruit fly (***Berdnik et al., 2006***; ***Sigg et al., 1997***), these results suggest that changes in GFP levels are likely due to changes in the promoter expression and not due to neuronal cell death.

In order to test whether expression of *Gr63a* and *Gr21a* is being downregulated by diacetyl, we used quantitative-PCR for testing along with a few members of the *Odorant receptor* (*Or*) family genes. Apart from *Gr63a* and *Gr21a,* we also evaluated receptors where diacetyl has no activity (*Or47a*, *Or88a*) and the broadly expressed co-receptor (*Or83b*). Wild-type flies that were exposed to diacetyl for 6 d showed dramatic downregulation of all genes tested. However, after a 5 d subsequent recovery period, expression of all but one (*Or88a*) of the genes was recovered, including *Gr63a* (***Figure 1—figure supplement 1C***). This form of plasticity in gene expression is unexpected and therefore mechanistically interesting to investigate.

## Diacetyl and structurally related odorants act as HDAC inhibitors *in vitro*

Diacetyl is structurally related to a soluble metabolite secreted by the gut microbiome and liver, β-hydroxybutyrate, which is known to inhibit zinc-dependent HDACs (***Shimazu et al., 2013***). It is also structurally related to another known HDAC inhibitor sodium butyrate. Diacetyl is present widely in nature as a pH-neutral fermentation product of microorganisms such as yeasts in over-ripe fruit, the favorite food for *Drosophila*, and also from lactic acid bacteria, and is found in many foods and beverages (***Shibamoto, 2014***). In humans, diacetyl is also produced by microbes on the skin, in the oral cavity, and is found in breath (***Whiteson et al., 2014***). Interestingly, this volatile is known to traverse the cell membrane (***Krogerus and Gibson, 2013***).

Soluble metabolites from the gut microbiome, such as short-chain fatty acids, are known to affect histone-modifying enzymes, gene expression, and physiology in various parts of the host (***McFall-Ngai et al., 2013***; ***Krautkramer et al., 2017***). It is presumed that these dissolved compounds absorbed in the blood circulate to various parts of the body. Microbes in our skin microbiome, mouth microbiome, and environment also emit a diverse repertoire of volatile metabolites that can traverse much greater distances away from the source (***James et al., 2013***). To examine whether diacetyl can directly modulate HDACs like β-hydroxybutyrate, we performed *in vitro* acetylation assays with purified recombinant HDACs. We found that, indeed, diacetyl inhibited all four purified human class I HDACs (HDAC1, 2, 3, and 8) and the one class II HDAC (HDAC6) tested. The inhibition occurred in a dose-dependent manner. The $IC_{50}$ values for HDAC1, 2, 3, 8, and 6 were 7.3, 23.1, 7.5, 14.3, and 24.5 mM, respectively (***Figure 1C***). The levels of inhibition as observed from $IC_{50}$ values for HDAC1 and HDAC3 were comparable to those of β-hydroxybutyrate whose $IC_{50}$ values were previously reported as 5.3 and 2.4 mM, respectively (***Shimazu et al., 2013***).

To test other odorants for their ability to inhibit HDACs, we identified a list of volatile odorants structurally similar to diacetyl using the similarity search feature on PubChem (***Figure 1D***) and tested them on four purified human HDACs, from class I (HDAC1 and HDAC3) and class II (HDAC4 and HDAC6). Each compound was tested at 15 mM, the concentration at which diacetyl inhibits HDAC1, 3, and 6 with more than 50% efficacy (***Figure 1C and D***). All compounds tested reduced the activity of at least one of the HDACs tested (***Figure 1D***). Ethyl pyruvate, methyl pyruvate, 2,3-hexanedione,

2,3-heptandione, 2,3-pentandione inhibited HDAC1, 3, and 6 with more than 70% efficacy. Allyl butyrate also inhibited those HDACs but with lower efficacy. 1-Acetoxyacetone and 2,3-butanediol inhibited HDAC6 more than 70% but inhibitory effects against HDAC1 and 4 were lower. Moreover, propyl formate strongly inhibited HDAC 6 but did not inhibit HDAC1, 3, and 4. Taken together, these results indicate that microbial volatiles and structurally related odorants can inhibit human HDACs, and that each compound has a specific HDAC inhibitory spectrum.

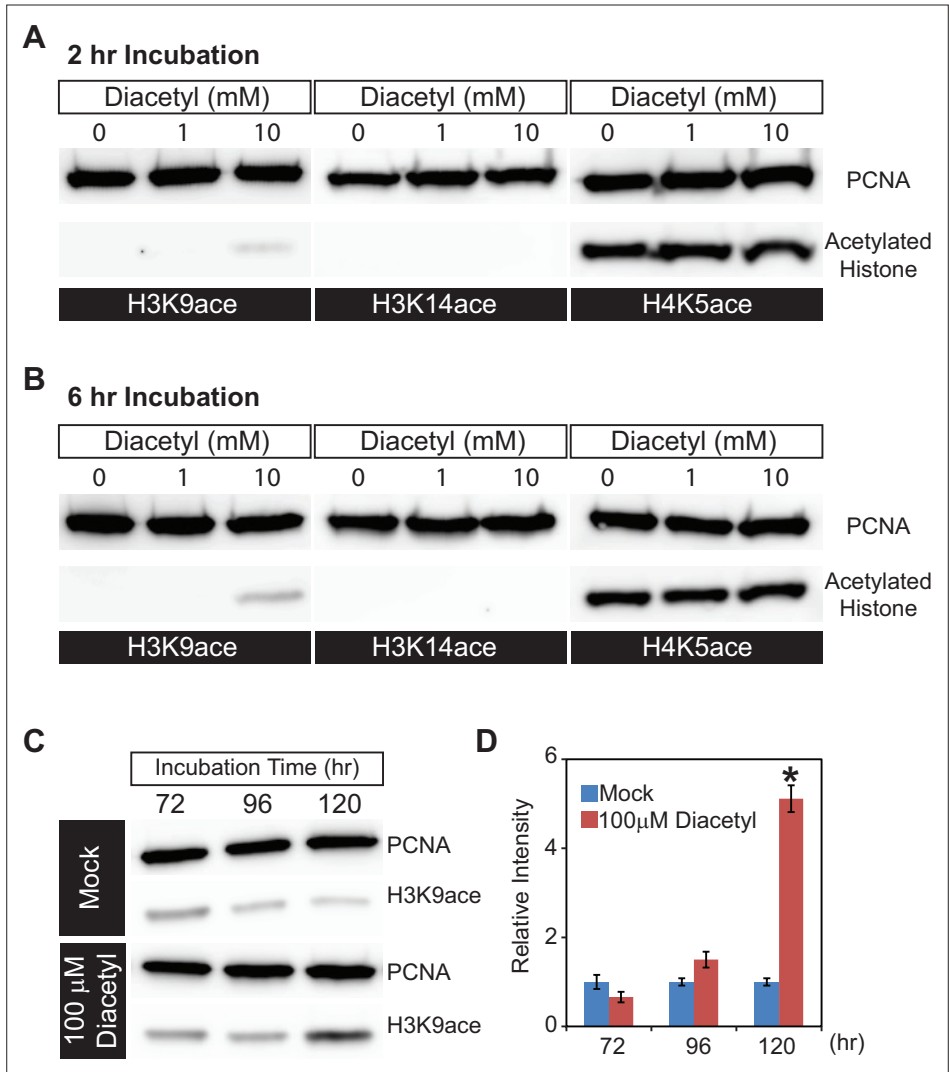

**Figure 2.** Diacetyl increases level of histone acetylation in HEK293 cells. (**A, B**) Representative images from western blots showing acetylation levels of H3K9 (left), H3K14 (middle), and H4K5 (right) in HEK293 cells after 2 hr (**A**) and 6 hr (**B**) of diacetyl exposure. Proliferating cell nuclear antigen (PCNA ) is a 29 kDa nuclear protein used as a loading control for nuclear protein extracts. (**C**) Western blots showing acetylation levels of H3K9 in HEK293T cells treated with 100 µM diacetyl for 72, 96, and 120 hr. PCNA is used for a loading control. (**D**) Bar graph showing the relative intensities of acetylated H3K9 in HEK293T cells treated with 100 µM diacetyl for 72–120 hr. n = 4 samples, * p<0.05.

The online version of this article includes the following source data for figure 2:

**Source data 1.** Original file for the Western blot analysis in *Figure 2A and B* (anti-H3K9ace).

**Source data 2.** Original file for the western blot analysis in *Figure 2A and B* (anti-H3K14ace).

**Source data 3.** Original file for the western blot analysis in *Figure 2A and B* (anti-H4K5ace).

**Source data 4.** Original file for the western blot analysis in *Figure 2C* (anti-H3K9AC).

## Microbial volatile increases histone acetylation in human cell nuclei

In order to directly examine histone acetylation at the cellular level, we used human HEK293T cells, which offer a tractable system to prepare nuclear extracts, and picked the volatile diacetyl to test in more detail. We exposed the cells to different doses of diacetyl for 2 or 6 hr and monitored histone acetylation levels within nuclear extracts in western blot assays. Compared to the mock treatment, 10 mM diacetyl significantly increased H3K9 acetylation levels within 2 hr of treatment, whereas the acetylation levels of H3K14 and H4K5 were not affected (*Figure 2A*). This specificity for an increase of the H3K9 mark is consistent with previous observations for β-hydroxybutyrate. Preferential acetylation of H3K9 was also observed previously in HEK293T cells with treatment of the structurally related β-hydroxybutyrate (*Shimazu et al., 2013*). After 6 hr of treatment, the H3K9 acetylation induced by 10 mM diacetyl was further increased (*Figure 2B*). The increase in H3K9 acetylation with diacetyl treatment was dependent both on the duration of exposure and concentration of the odorant.

Organisms are constantly exposed to volatiles commonly found in their food and environment for prolonged periods of time. In order to test the effect of a lower concentration of the volatiles, we selected a 5 d exposure time at a 100-fold lower concentration. When we treated HEK293T cells with this lower dose of diacetyl (100 µM), H3K9 acetylation level increased after 96 hr of exposure and reached significantly higher levels than control after 120 hr (*Figure 2C and D*). These results demonstrate that even prolonged exposure to low levels of diacetyl can greatly impact the epigenetic landscape inside the nucleus. More importantly, a 5 d exposure was sufficient to alter the epigenetic state of nuclei at concentrations that are present in some food sources. Taken together, these results demonstrate that diacetyl can act as an HDAC inhibitor, with the potential of causing broad modulations of gene expression and histone acetylation in cells.

## Transcriptional response to odor exposure is conserved in invertebrates and vertebrates

We next performed *in vivo* experiments to determine whether eukaryotes alter gene expression in response to diacetyl exposure, as would be expected with HDAC inhibition. Epigenetic changes occur slowly over days, and to test its effects on gene expression we used two model systems for eukaryotes: invertebrates (*D. melanogaster*) and vertebrates (*M. musculus*). We placed *Drosophila* adult males in vials and housed these vials within a closed container exposed to headspace from a 1% diacetyl solution (V/V in paraffin oil) for 5 d, similar to a previous long-term odor-exposure study (*Sachse et al., 2007*; *Figure 3A*). The transcriptome of the primary olfactory organ, the antenna, was compared with that of the control group of age-matched flies that were exposed to the solvent paraffin oil (PO). The antennal transcriptional profile of diacetyl-exposed flies showed substantial changes in gene expression when compared to the solvent control (*Figure 3B*). We identified 1234 differentially expressed genes (DEGs) (false discovery rate [FDR] < 0.05) in the antennal transcriptome of diacetyl-exposed flies compared to control animals. Of these, 645 genes were significantly upregulated ($\log_2$ fold-change > 1; red dots in *Figure 3B*) and 589 genes were significantly downregulated ($\log_2$ fold-change < –1; blue dots in *Figure 3B*). A broad range of genes were significantly altered, with several biological process GO terms significantly enriched in the upregulated gene list including 'response to biotic stimulus' ($p<3 \times 10^{-7}$), 'response to bacterium' ($p<3 \times 10^{-7}$), and 'defense response' ($p<3 \times 10^{-7}$). In the downregulated gene set, the GO term most enriched was 'sensory perception of chemical stimulus' ($p<6 \times 10^{-72}$). The GO molecular functions that were most common in the upregulated sets included enzymes like hydrolases and oxidoreductases, as well as nucleic acid binding genes (*Figure 3C*). Downregulated genes include receptors, hydrolases, and transporters (*Figure 3C*).

The first tissue that would have contact with airborne volatiles in terrestrial vertebrates are the airways and lungs. We therefore performed transcriptome analyses on lung tissue of mice exposed to diacetyl headspace at different doses for a period of 5 d, as was done in *Drosophila* (*Figure 3D*). The dose of the volatile in the chamber was measured in parallel experiments by tracking volatile loss and airflow and determined to range between a peak 1920 ppm at the end of hour 2, tapering off to 28 ppm by hour 120 (*Figure 3—figure supplement 1B*). Indeed, expression of a substantial number of genes was modulated in the diacetyl-exposed lungs compared to the control. The changes were dose-dependent and more pronounced in mice exposed to headspace from a 1% diacetyl source compared to those exposed similarly to 0.1% (v/v) diacetyl source (*Figure 3E*, *Figure 3—figure supplement 1C*). Among these diverse sets of regulated genes in the lung tissue, a significant overlap

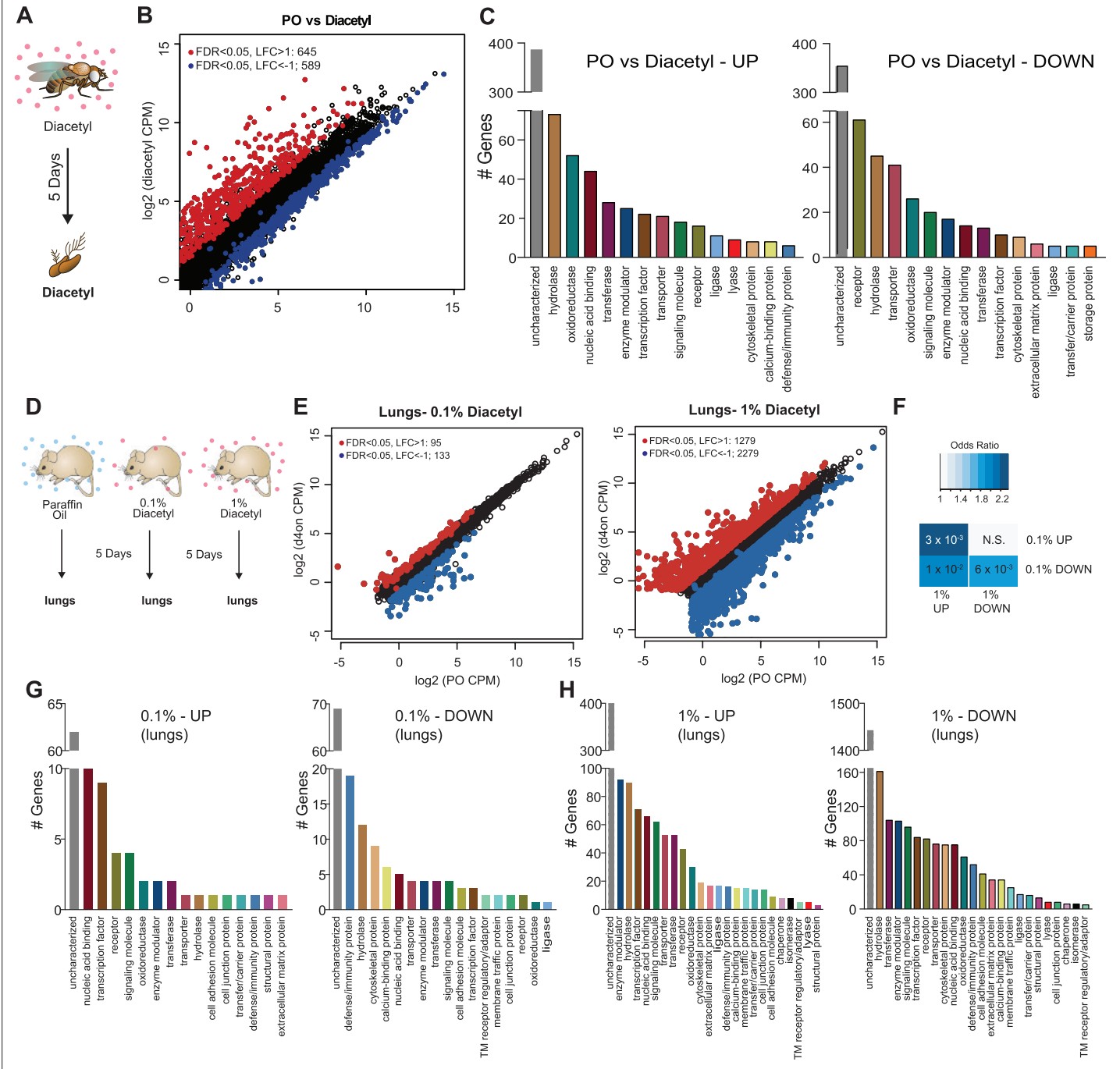

**Figure 3.** Remote exposure to HDACi volatile alters gene expression in an invertebrate and vertebrate. (**A**) Schematic of odor exposure protocol for transcriptome analysis from the antennae. (**B**) Plot highlighting up- and downregulated genes in the diacetyl-exposed group. Red and blue dots represent upregulated genes (false discovery rate [FDR] < 0.05, $\log_2$ fold change [LFC] > 1) and downregulated genes (FDR < 0.05, LFC < 1), respectively. (**C**) Bar graphs denoting the protein classification of the genes up- and downregulated after odor exposure. (**D**) Schematic of diacetyl exposure protocol for transcriptome analysis of mouse lung tissue. (**E**) Plot highlighting up- and downregulated genes in the diacetyl-exposed groups. Red and blue dots represent upregulated genes (FDR < 0.05, LFC > 1) and downregulated genes (FDR < 0.05, LFC < 1), respectively in lungs. (**F**) Table showing pairwise tests of significance of overlap between gene sets. p-Values from Fisher's exact test, colored with associated odds ratio (strength of association). (**G, H**) Bar graphs denoting the protein classification of the genes up- and downregulated in the lung after exposure to headspace above 0.1% (v/v) (**G**) and 1% (v/v) (**H**) of diacetyl solutions.

The online version of this article includes the following source data and figure supplement(s) for figure 3:

**Source data 1.** Concentration of diacetyl in some common sources.

**Figure supplement 1.** Dose of diacetyl in ppm in experimental chambers over time.

was found between 1% (v/v) and 0.1% (v/v) headspace for both upregulated genes (p=3 × 10⁻³) and downregulated genes (p=6 × 10⁻³, *Figure 3F*), further supporting a dose-dependent effect of diacetyl on gene expression in the mouse lung.

The GO enrichment analysis of the lung transcriptome from mice exposed to diacetyl revealed several interesting sets of genes that were significantly altered (*Figure 3G and H*). Among these genes is the SARS-CoV-2 entry receptor ortholog, ACE2. The level of this gene in the lungs of diacetyl-exposed mice showed a reduction in expression levels ($\log_2$ fold-change = - 0.62, FDR = 2.02 × 10⁻⁵). The level of ACE2 in the nasal tissue, lungs, and other organs plays a critical role in infection by the SARS-CoV-2 virus, suggesting that epigenetic mechanisms may contribute to expression. While we do not know whether expression of ACE2 in humans is regulated similarly, these results highlight the importance of understanding volatile-induced modulation of gene expression in the lungs and other tissues.

HDAC family members are highly conserved across eukaryotes, spanning both animal and insects. Volatile microbial metabolites have been present throughout the evolution of animals, insects, and plants, and have potential as signals for multidomain communication because they can travel wide ranges; plants detect root infections and time their immune response using volatiles (*Effantin et al., 2011*), and microbial volatiles elicit olfactory behaviors in insects and nematodes. Our results indicate that a volatile compound, capable of directly inhibiting HDACs and altering acetylation levels, can cause significant changes in gene expression in a variety of organisms following exposure from a distance, through the air.

## DEGs overlap those for known HDAC inhibitors and upregulation is partially reversible

One of the hallmarks of epigenetic regulators which makes them attractive candidates for therapeutics is plasticity. Upon removal of the inhibitors, we expect that some of the changes in gene expression are reversible. To test for reversibility, we performed a recovery experiment following diacetyl exposure using the *Drosophila* model. We maintained 5 d diacetyl-exposed flies in clean air for five additional days (*Figure 4A*). In parallel, we performed age-matched mock experiments with paraffin oil solvent exposure alone. A large number of genes were downregulated following the recovery in comparison to the 10-day-old flies in the mock condition (*Figure 4B*). Interestingly, there was a significant overlap of these downregulated genes with the set that was upregulated in the diacetyl treatment (*Figure 4C*). These results suggest that the effects of HDAC inhibitory odorant exposure are not permanent but dynamic, and removal of the odorant leads to subsequent changes in gene expression of the upregulated set.

Drugs that inhibit HDACs are in various stages of approval as treatments for several diseases. The approved drugs include sodium butyrate and valproic acid. To compare the gene-regulatory effects of these drugs to diacetyl, we performed RNA-seq after raising the flies on food containing sodium butyrate (SB) or valproic acid (VA) (*Steffan et al., 2001*) and compared gene expression in the antennae of flies raised on untreated food for 5 d. We next compared the upregulated gene profiles following each treatment to the one induced by exposure to diacetyl. As expected, feeding SB and VA induced significant changes in expression levels of several genes (*Figure 4D and E*). Interestingly, we found that 113 of diacetyl upregulated genes were also upregulated in either SB, VA, or both treatment conditions (*Figure 4F*). Pairwise statistical analysis of each gene set revealed a significant overlap of diacetyl-induced genes with SB-induced genes (p=6 × 10⁻¹¹) and with VA-induced genes (p=2 × 10⁻⁶⁵) (*Figure 4F*). There was, as expected, also a significant overlap between SB- and VA- induced genes (p=1 × 10⁻⁵²) (*Figure 4F*). This highly significant overlap among upregulated genes lends further support to our model that diacetyl vapors act as an HDAC inhibitor *in vivo*. As expected, each of the three treatments also modulated a substantial number of unique genes (*Figure 4G and H*), suggesting that differences in oral vs. vapor delivery, molecular structure, and inhibition profile across the repertoire of HDACs may contribute to differences in gene regulation. These results suggest that some volatile odorants, or the microbes that emit them, may impart therapeutic effects like other HDAC inhibitors from a distance, through the air.

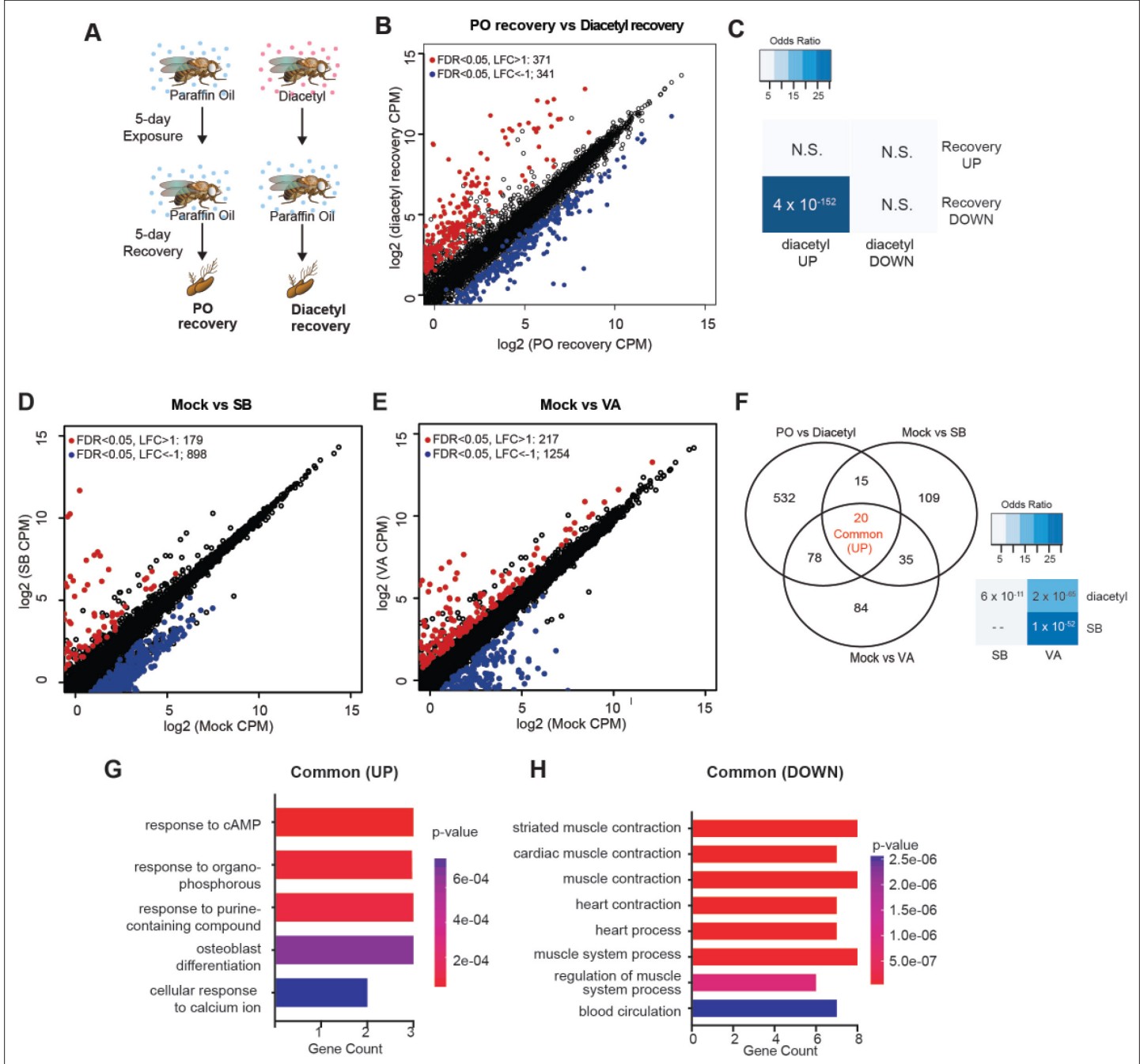

**Figure 4.** Gene expression changes are partly reversible and overlap with known HDACi drugs. (**A**) Schematic of odor exposure and recovery protocol for transcriptome analysis from the antennae. (**B**) Plot highlighting up- and downregulated genes in the recovery from diacetyl exposure group. Red and blue dots represent upregulated genes (false discovery rate [FDR] <0.05, $\log_2$ fold change [LFC] > 1) and downregulated genes (FDR < 0.05, LFC < 1), respectively. (**C**) Table showing pairwise tests of significance of overlap between gene sets. p-vVlues from Fisher's exact test, colored with associated odds ratio (strength of association). (**D, E**) Plots showing enrichment of up- and downregulated genes in sodium butyrate- (**D**) and valproic acid-treated (**E**) groups. Red and blue dots represent upregulated genes (FDR < 0.05, LFC > 1) and downregulated genes (FDR < 0.05, LFC < 1), respectively. (**F**) Left: Venn diagrams showing the overlaps of upregulated genes among diacetyl-, sodium butyrate-, and valproic acid-treated groups. Right: table showing pairwise tests of significance of overlap between gene sets. p-Values from Fisher's exact test. (**G, H**) GO enrichment analysis of the common upregulated and downregulated genes across all three treatments in (**F**).

## Volatile diacetyl alters gene expression in the mouse brain

One of the categories of challenging diseases that HDAC inhibitors are being tested for drugs are neurodegenerative diseases (*Fischer, 2010*), where volatile odorants may provide a new class of inhaled therapeutics. A major design challenge for nervous system drugs is the ability to cross the blood–brain barrier. Due to the volatility and small size of volatile odorants, there is a possibility that they could diffuse through the intranasal route to the brain directly (*Chauhan and Chauhan, 2015*). We cannot tell a priori whether odorants like diacetyl can travel across the nasal membrane to the brain. In order to test whether cells in the brain of mice respond when presented with diacetyl vapors from a distance, we used gene expression as the final readout to measure it. We performed RNA-seq experiments on mice exposed only to aroma of diacetyl in the air for 5 d as before. Littermate controls were exposed in a similar manner to the solvent (PO) in the air. Interestingly, several genes were differentially expressed upon exposure to headspace from 0.1% (v/v) diacetyl passaged through the air in the holding cage where it is further diluted (49 upregulated, 32 downregulated, $|\log_2$ fold-change$|>1$, FDR < 0.05) or to headspace from 1% (v/v) diacetyl diluted in the holding cage (748 upregulated, 1031 downregulated, $|\log_2$ fold-change$|> 1$, FDR < 0.05) (*Figure 5B and C*). GO analysis of the regulated genes in the exposed mouse brain transcriptome revealed several interesting sets of genes were significantly altered in each set (*Figure 5E*). These results indicate that a volatile odorant like diacetyl can reach the brain in mammals and presumably alter gene expression in neuronal cells. Although the overall DEG sets were different across the lungs and brain, there was a statistically significant overlap, particularly in the 1% exposure, as would be expected from using the same mechanism of action. There was a highly significant overlap ($p=2 \times 10^{-101}$) between genes upregulated in the brain compared with the lungs (248 of 748), as well as for downregulated genes ($p=3 \times 10^{-163}$; 477 of 1031 genes) (*Figure 5D*).

## Diacetyl prevents proliferation of neuroblastoma cells

Among the significantly downregulated genes was *MYCN*, which is known to show increased expression and plays a central role in many types of cancers, suggesting that epigenetic regulation may affect tumor cells. Broad-spectrum HDAC inhibitors have been approved by the FDA for treatment of a variety of cancers. To test the exciting possibility that a volatile HDAC inhibitor, which affects gene expression in the brain, would have an effect against cancers as well, we performed a cell-based assay. We picked neuroblastoma as a candidate since it has a strong dependence on *MYCN* and develop from fetal adrenal neuroblasts (*Jansky et al., 2021*). Several genetic and 'omics' studies have identified genes that are upregulated in neuroblastomas, some of which are considered to be key causative factors (*Pugh et al., 2013*). We evaluated the DEGs from the diacetyl-exposed mouse brain transcriptome for several of these key genes. Three of the six key genes identified across multiple studies were significantly altered, two of which were significantly downregulated, *MYCN* ($p=2.06E-13$, FDR = 1.76E-12) and *Alk* ($p=3.3E-5$, FDR = 9.48E-5) (*Figure 5F*). An unbiased computational study evaluated omics datasets and identified four genes that were the best predictors of clinical outcome (*Zhong et al., 2018*), all of which are significantly different in the brains of the diacetyl-exposed mice, strikingly three of which were significantly downregulated, *NCAN* ($p=1.33E-16$, FDR = 1.59E-15), *STK33* ($p=4.6E-7$, FDR = 1.79E-6), and *ERCC6L2* ($p=4.57E-5$, FDR = 0.00012) (*Figure 5F*). These findings are extremely provocative, given that the downregulated genes, particularly MYCN, play a role in neuroblastomas and in several other cancers.

We tested diacetyl for the ability to inhibit proliferation of an SH-SY5Y neuroblastoma cell line from a metastatic bone tumor. A significant reduction in cell count was seen in diacetyl treatments in the SH-SY5Y cancer cell lines (*Figure 5G*). To validate the initial result, we retested diacetyl at two concentrations across three different cancer cell lines. Diacetyl showed a significant reduction in cell numbers for the neuroblastoma cell line specifically, and not any other cell lines, relative to the control (*Figure 5G*). These results demonstrate that the effect is specific to the SH-SY5Y cancer cell line and unlikely to be caused by a nonspecific general toxic effect. It also suggests that the potential epigenetic regulation of cancer-promoting genes may be caused by exposure to volatiles like diacetyl in the environment.

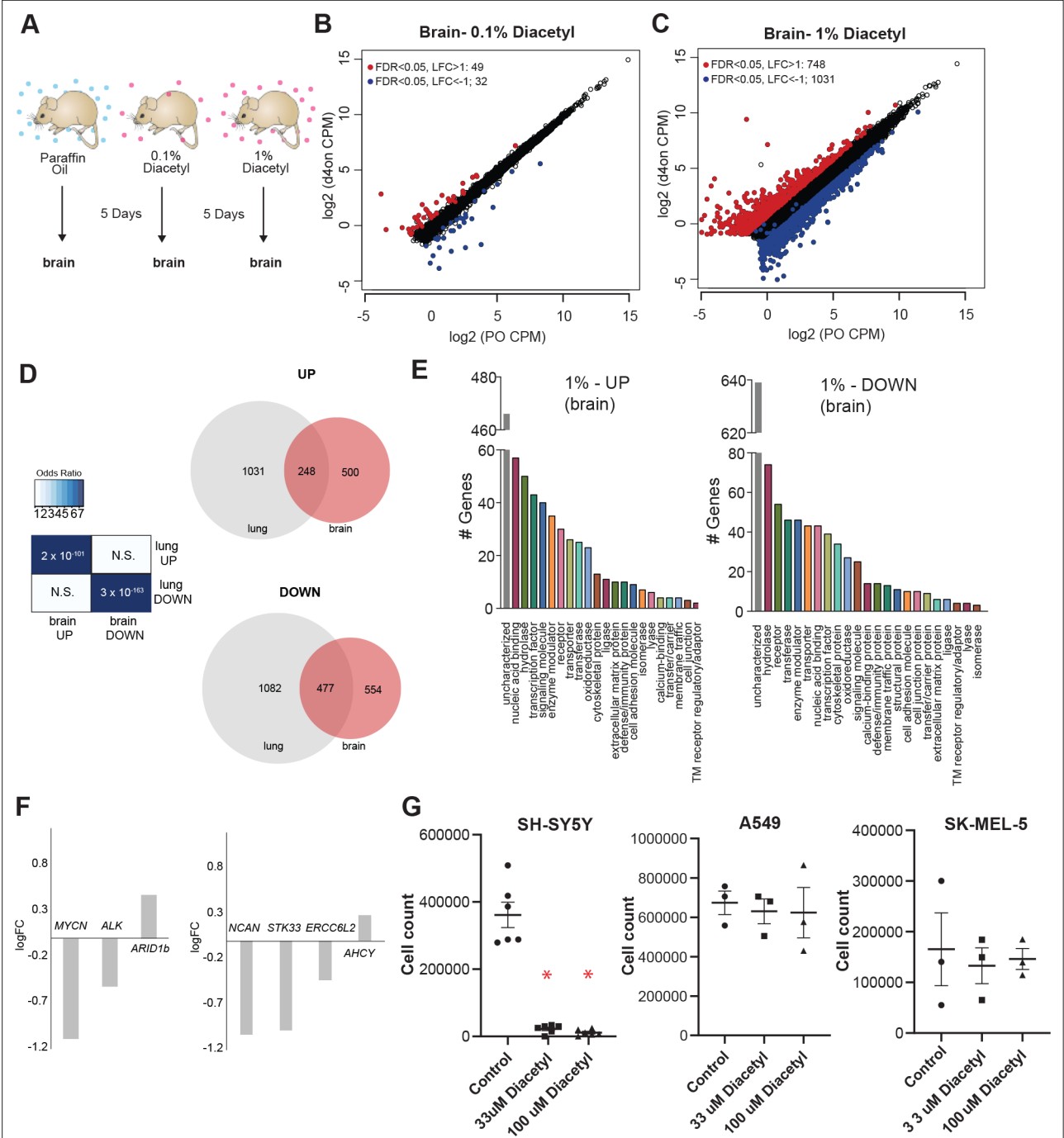

**Figure 5.** Exposure to diacetyl vapor alters gene expression in the mouse brain. (**A**) Schematic of diacetyl exposure protocol for transcriptome analysis of mouse brain tissues. (**B, C**) Plot showing up- and downregulated genes in the diacetyl-exposed groups. Red and blue dots represent upregulated genes (false discovery rate [FDR] < 0.05, $\log_2$ fold change [LFC] > 1) and downregulated genes (FDR < 0.05, LFC < 1), respectively, in the brain. (**D**) Left: table showing pairwise tests of significance of overlap between gene sets. p-Values from Fisher's exact test. Right: Venn diagrams showing the overlaps of differentially expressed genes (DEGs) between lung and brain groups. (**E**) Bar graphs denoting the protein classification of the brain genes up- and downregulated after 1% diacetyl exposure. (**F**) Mean fold change of select neuroblastoma related genes in the brain of mice exposed to diacetyl vapors. (**G**) Cell counts of indicated cancer cell lines in tissue culture treated with solvent control or indicated concentration of diacetyl. N = 3–6, p<0.001.

## Testing therapeutic effect in Huntington's neurodegeneration model

Since diacetyl can affect changes in a mammalian brain by crossing the blood–brain barrier, we wanted to test whether it can impart therapeutic effects in disorders of the brain like neurodegeneration for which HDAC inhibitors have been proposed as therapeutics. In order to test diacetyl in the context of neurodegenerative disease, we used the well-established *Drosophila* model of Huntington's disease, which has previously been used to demonstrate the efficacy of sodium butyrate in slowing neurodegeneration (*Steffan et al., 2001*; *Marsh and Thompson, 2006*). The human Huntingtin protein with expanded poly-Q repeats is expressed in the neurons of the compound eye in *Drosophila*, causing progressive degeneration of the seven visible photoreceptor rhabdomere cells in each ommatidium (*Jackson et al., 1998*). We selected this model in particular since polyglutamine disorders are well suited for targeting by inhibitors of HDAC1 and HDAC3 (*Thomas, 2014*) such as diacetyl (*Figure 1B*). Moreover, previous studies have shown that orally administered HDAC inhibitors such as sodium butyrate and SAHA can significantly reduce photoreceptor degeneration in this model (*Steffan et al., 2001*). When the transgenic flies expressing two copies of the human Huntingtin with poly-Q repeats (HTTQ120) under control of the eye-specific GMR promoter were raised at 18°C, the number of rhabdomeres in each ommatidium was similar to that normally observed in wild-type flies immediately post-eclosion (day 1, *Figure 6A–C*). When these HTTQ120 flies were moved to 25°C following eclosion (*Figure 6A*), they showed dramatic degeneration of rhabdomeres over a period of 10 d (*Figure 6B and D–G*). The mean number of rhabdomeres was reduced from 7 to ~1 by day 10. The dose of the volatile in the chamber was measured in parallel experiments by tracking volatile loss and airflow (0.5 L/min) and determined to range from 260 to 95 ppm, an average of 125 ppm at the end of hour 1, tapering off to 0 ppm by hour 24. The odorant was replaced each day, leading to a dosage profile that would seem like a once-a-day treatment regimen (*Figure 3—figure supplement 1A*).

Remarkably, when the Huntingtin (HTTQ120)-expressing flies were exposed immediately after eclosion to volatile headspace of 1% diacetyl (*Figure 6A*), they showed a substantial (~50%) reduction of rhabdomere loss (*Figure 6B and D–G*). The majority of ommatidia retained 6–7 rhabdomeres at day 5 (*Figure 6D and F*). Even after 10 d, the majority of the ommatidia still had 2–3 rhabdomeres left in the odor-exposed flies, while the solvent controls had only ~1–2 rhabdomeres (*Figure 6E and G*). These results demonstrate that exposure to volatile diacetyl in air slows down the photoreceptor degeneration caused in these Huntington's disease model flies. Thus, odorants like diacetyl may have potential as a prophylactic against this neurodegenerative disorder. Therefore, the volatile odorant can both significantly alter gene expression from a distance as well as cause a phenotypic change in the fly with a human disease gene.

## Discussion

In this study, we discovered that eukaryotic cells could alter gene expression in response to an odorant in a manner independent of traditional neuronal activity-induced pathways. Members of a conserved family of HDACs detected the concentration of diacetyl differentially. Diacetyl exposure increased levels of H3K9 acetylation in the nucleus in a dose-dependent manner and led to changes in gene expression. The odorant diacetyl occurs naturally and is generated from the metabolism of a variety of food components, including triglycerides, sugars, and amino acids (*Shibamoto, 2014*). Moreover, the chemical is produced by the activity of microorganisms such as yeasts and lactic bacteria during fermentation in many foods and beverages (*Shibamoto, 2014*; *Figure 3—source data 1*). Although beyond the scope of this study, the discovery that cells can alter gene expression upon prolonged exposure to a common naturally occurring odorant raises important questions about the potential physiological consequences of the expression changes. Given our repeated exposure to particular flavors and fragrances, the findings outlined here highlight a new consideration for evaluating the safety of certain volatile chemicals that can cross the cell membrane.

In mammals, upregulation of circulating β-hydroxybutyrate during fasting or calorie restriction induces changes in the expression of a set of genes (*Shimazu et al., 2013*). The acetylation mark specificity and $IC_{50}$ of β-hydroxybutyrate is similar to that of diacetyl. Any beneficial physiological effects of odor detection at lower odor concentrations would also be countered by studies of potential risks at higher concentrations. In fact, the deleterious effect of exposure to high levels to diacetyl causing bronchiolitis obliterans, or 'popcorn lung,' and its toxicity in cultured cells is already known

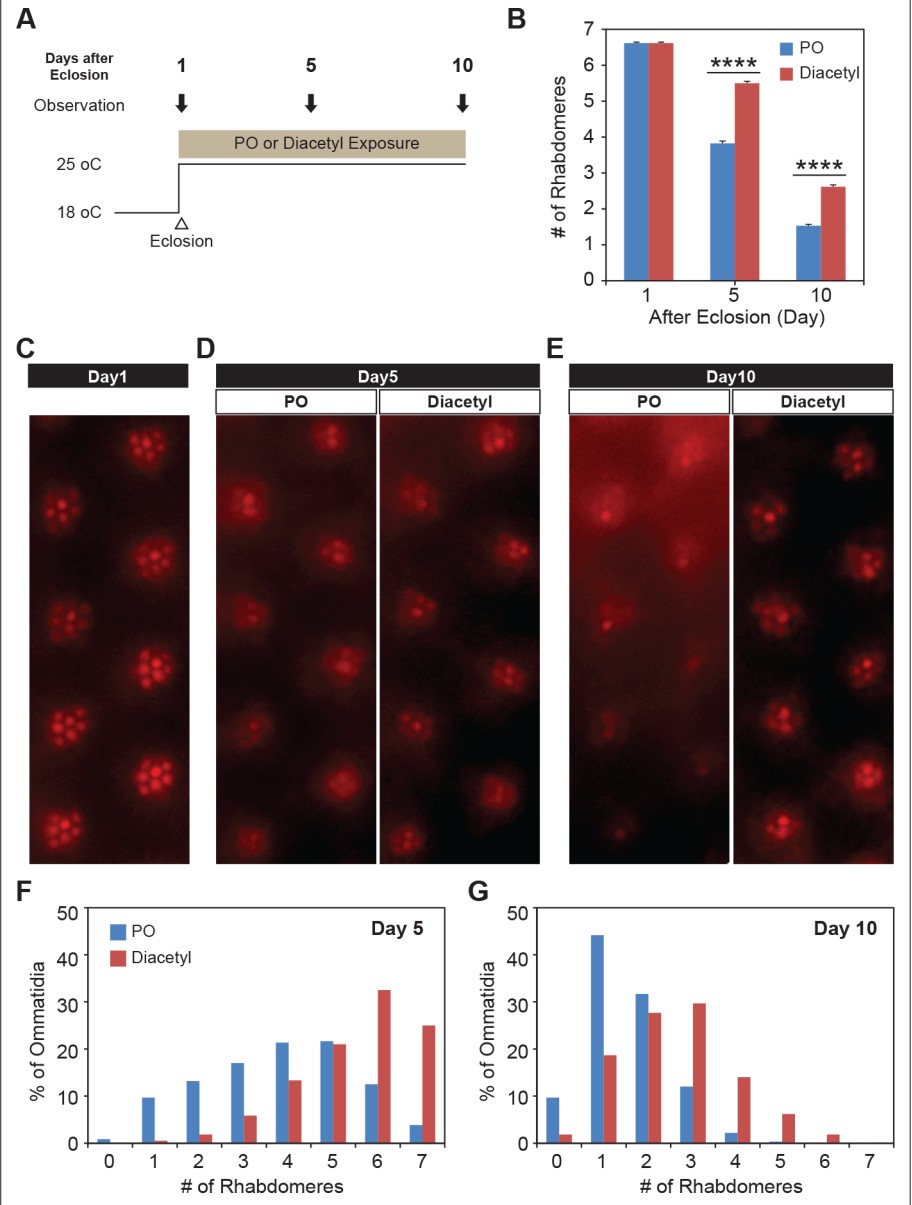

**Figure 6.** Odor exposure slows Huntington's neurodegeneration model in fly eye. (**A**) Schematic diagram showing temperature of experimental condition and timing of the eye examination in pGMR-HTTQ120 flies. (**B**) Bar graph showing mean number of rhabdomeres in each ommatidium in solvent paraffin oil (PO, blue) and diacetyl-exposed (red) pGMR-HTTQ120 flies at 1, 5, and 10 d after eclosion (AE). n = 600 ommatidia from 15 flies, ****p<0.0001. (**C**) A representative image of ommatidia of pGMR-HTTQ120 flies at 1 d AE. (**D, E**) Representative images of ommatidia of pGMR-HTTQ120 flies exposed to paraffin oil (PO) or diacetyl at 5 (**D**) and 10 (**E**) d AE. Scale bars, 5 μm (**C, D, E**). (**F, G**) Histogram showing the percent of the ommatidium with a given number of rhabdomeres indicated on the x axis at 5 (**F**) and 10 (**G**) d AE.

(***More et al., 2012***). Even so it is present in several foods we eat and is on the Generally Regarded as Safe list of Flavor and Extract Manufacturers Association for use as a flavoring ingredient at low concentrations. While the specific mechanism of 'popcorn lung' is unknown, it has previously been proposed that carbonyl groups of diacetyl can react to modify amino acid side chains on proteins or generate reactive dicarbonyl and reactive oxygen species, leading to excessive cytokine production and inflammation (***Starek-Swiechowicz and Starek, 2014***). Our results raise the possibility that the molecular mechanisms underlying this disorder could also be partially attributed to the unusually high HDAC-mediated genetic response to diacetyl by cells in the lungs. Several studies using rodent

animal models have shown toxicity of high levels of diacetyl in the respiratory tracts including the nose and lungs (*Morgan et al., 2008*; *Palmer et al., 2011*). The toxic effect of higher dosages has been observed for many types of HDAC inhibitors. For example, in one *Drosophila* study, feeding of a high dosage of 4-phenylbutyrate reduces survival rate, while a lower dosage extends longevity (*Kang et al., 2002*). Another concern is that one of the highest known levels of diacetyl exposure to humans is through smoking cigarettes, where exposure from even a single cigarette (250–361 ppm) is high (*Rigler and Longo, 2010*; *Pierce et al., 2014*): Salem (128 μg/cigarette), Camel (128.1 μg/cigarette), and Carlton (45.6 μg/cigarette) (*Pierce et al., 2014*). Different types of diets and lifestyle choices have been known to have effects on health and well-being due to differences in balance of different health-promoting ingredients, yet little is known about the role of such flavors, fragrances, and metabolites acting on epigenetic marks and altering gene expression. Our findings raise the possibility that diacetyl and structurally related odors in the environment may affect gene expression directly through absorption/transport into these cells. In mammals, the epithelium-lined lungs and nasal/oral pathway is directly exposed to odorants, presenting an opportunity to affect gene expression and physiology.

Eukaryotes primarily detect volatile odorants in the environment using olfactory neurons that express a variety of transmembrane receptors, a family of genes that have independently evolved multiple times in different phyla. Examples of these unrelated genes include the ionotropic 7-transmembrane (TM) odorant receptor (Or) family, which is insect-specific; 3-TM ionotropic receptors, which are present across most arthropods; the nematode-divergent 7-TM GPCRs belonging to the *str, sra, srg, srw, srz, srbc, srsx,* and *srr* families; the mammalian 7-TM GPCR olfactory receptor (OR) family; and the trace amine-associated receptor family (*Buck and Axel, 1991*; *Clyne et al., 1999*; *Troemel et al., 1995*; *Benton et al., 2009*; *Vosshall et al., 1999*). The activation of these receptors induces neuronal action potentials, and this information is conveyed to higher brain centers where olfactory perception is generated (*Knaden and Hansson, 2014*). This second response pathway that we have discovered is much slower and has an enzymatic mechanism that draws parallels with the multiple pathways in place to detect light. In the specialized cells of the eye, detection of light occurs via rhodopsins (7-transmembrane GPCRs) and leads to neuronal activity and behavioral responses. However, other cells, including in plants, are also able to respond to changes in light intensity of certain wavelengths using the ancient, conserved cryptochrome proteins that are related to photolyases (*Fogle et al., 2015*). This suggests that vital sensory modalities can be detected via multiple pathways. Our observations raise the question whether HDACs represent a conserved detection mechanism for odorants like diacetyl that link odor detection to a specific gene expression response. HDAC proteins, however, are an ancient family of genes that predate even histones themselves (*Gregoretti et al., 2004*; *Postberg et al., 2010*). It is conceivable that odor detection mechanisms that emerged in ancient forms may not have involved specialized transmembrane receptors, or neurons, or resulted in rapid behavioral movements.

Our data suggests that an HDAC inhibitory odorant also affects gene expression in the lungs and brain of mice, including significantly decreasing the expression level of the ACE2 receptor, whose ortholog in humans is the entry receptor for the SARS-CoV-2 virus that causes COVID-19. The applied impact for the future could be far-reaching as HDACs are known to affect a wide variety of pathways in eukaryotes, and inhibitors are candidates for novel inhaled therapeutics as treatment of many conditions such as COVID-19, cancers, neurodegeneration, and a variety of infectious diseases (*Witt and Lindemann, 2009*; *Wang et al., 2013*; *Wang et al., 2012*; *Yeung et al., 2012*; *Park et al., 2012*; *Cai et al., 2010*; *Shankar and Srivastava, 2008*; *Platta et al., 2007*; *Komatsu et al., 2006*).

Our studies also show that one of the volatiles is protective against the neurodegenerative effects of Huntington's disease model and neuroblastoma. Exposure to diacetyl volatiles in the fly model of Huntington's disease reduces cell degeneration, as has been previously observed with orally administered HDAC inhibitors like sodium butyrate and SAHA in this genetic model (*Steffan et al., 2001*). Previous studies indicate that the inhibition of HDACs counters the acetyltransferase inhibitory activity of the polyglutamine-domain of the Htt protein, which binds to p300, P/CAF, and CBP (*Steffan et al., 2001*).

This newly discovered activity in this class of volatile chemicals will also allow us to understand how their presence in our environment, diet, and microbiome shapes the epigenetic landscape in the lungs and affects inflammation and immune responses. Our approach to investigate consequences in model systems allows for a foundation for future research and allow identification of key molecular

biomarkers that can be associated with observation of epigenetic effects, toxic or not, in individuals. The public health impact of these findings can be far-reaching, both from the perspective of toxicity and therapeutics. Investigating the health safety concerns of exposure to such odorants at high doses or prolonged periods is essential to understand potential for damage. Conversely, at nontoxic concentrations, HDAC inhibitors can be used for the treatment of many debilitating conditions in the lungs such as COPD, inflammation, and COVID-19 infection (*Shakespear et al., 2011*).

Taken together, our discovery that cells can alter gene expression in response to an odor that acts as an HDAC inhibitor and reprogram gene expression promises the pursuit of new types of odor-based therapeutics already approved for human consumption as prophylactics for a myriad of diseases. Simultaneously, they raise the concern of a new type of environmental agent that can reprogram gene expression and have widespread effects that are yet unknown.

# Materials and methods

**Key resources table**

| Reagent type (species) or resource | Designation | Source or reference | Identifiers | Additional information |
|---|---|---|---|---|
| Genetic reagent (*Drosophila melanogaster*) | P{GMR-HTT.Q120}2.4 | Bloomington Drosophila Stock Center | BDSC:8533 | |
| Genetic reagent (*D. melanogaster*) | Gr63a-Gal4 | Bloomington Drosophila Stock Center | BDCS:9943 | |
| Genetic reagent (*D. melanogaster*) | UAS-mcd8:GFP | *Lee and Luo, 1999* | mCD8-GFP in pUAST | Gift from Dr. John Carlson's lab, Yale |
| Genetic reagent (*D. melanogaster*) | Gr21a-Gal4 | Bloomington Drosophila Stock Center | BDSC:57600 | |
| Strain, strain background (*D. melanogaster*) | wCS | *Koh et al., 2014* | wCS | w1118 backcrossed multi-generation to Canton-S |
| Strain, strain background (*Mus musculus*) | C57BL/6J | The Jackson Laboratory | Stock no.: 000664 | |
| Cell line (human) | HEK293T | ATCC | CRL-3216 | Gift from Dr.Francis Sladek & Dr.Xuan Liu, UCR |
| Cell line (human) | A549 | ATCC | CCL-185TM | Lot # 70035208 |
| Cell line (human) | SK-MEL-5 | ATCC | HTB-70 | Gift from Dr. Maurizio Pellecchia Lab, UCR |
| Cell line (human) | SH-SY5Y | ATCC | CRL-2266 | Gift from Dr. Maurizio Pellecchia Lab, UCR |
| Antibody | Anti-nc82 (mouse monoclonal) | Development Studies Hybridoma Bank | nc82 | IF (1:20) |
| Antibody | Anti-GFP (rabbit polyclonal) | Invitrogen | A-11122 | IF (1:150) |
| Antibody | Anti-rabbit IgG, Alexa Fluor Plus 488 | Invitrogen | A32731 | IF (1:400) |
| Antibody | Anti-mouse IgG, Alexa Fluor 568 | Invitrogen | A-11004 | IF (1:400) |
| Antibody | Anti-acetyl Histone H3K9 (rabbit polyclonal) | abcam | ab4441 | WB (1:2000) |
| Antibody | Anti-acetyl Histone H3K14 (rabbit polyclonal) | EMD Millipore | 06-911 | WB (1:5000) |
| Antibody | Anti-acetyl Histone H4K5 (rabbit polyclonal) | EMD Millipore | 07-327 | WB (1:2000) |
| Antibody | Immun-Star anti-rabbit IgG, HRP (goat polyclonal) | Bio-Rad | 1705046 | WB (1:20,000) |
| Sequence-based reagent | This paper | Gr21a F | PCR primer | CGATCGTCTTTCCGAATCTC |

*Continued on next page*

*Continued*

| Reagent type (species) or resource | Designation | Source or reference | Identifiers | Additional information |
|---|---|---|---|---|
| Sequence-based reagent | This paper | Gr21a R | PCR primer | GGCTCAGATCCACCCATAGA |
| Sequence-based reagent | This paper | Gr63a F | PCR primer | AAATGAACTCCGCCTCCTTT |
| Sequence-based reagent | This paper | Gr63a R | PCR primer | CGCAATTTCAGAGGCAAACT |
| Sequence-based reagent | This paper | RP49 F | PCR primer | CTGCCCACCGGATTCAAG |
| Sequence-based reagent | This paper | RP49 R | PCR primer | GTTTCATGCGGCGAGATCG |
| Sequence-based reagent | This paper | Or47a F | PCR primer | ATCACAGGCCACATTGAACA |
| Sequence-based reagent | This paper | Or47a R | PCR primer | TCCCCGCAGTAGCAGTAGAT |
| Sequence-based reagent | This paper | Or88a F | PCR primer | TTAAAGTGGCCTTCCTGGTG |
| Sequence-based reagent | This paper | Or88a R | PCR primer | ATGCGGCAATAAAGTTCCAC |
| Sequence-based reagent | This paper | Or83b F | PCR primer | TTCTTGGCATTCGCTTTTCT |
| Sequence-based reagent | This paper | Or83b R | PCR primer | TCCCTGGATTTGTTTGCTTC |
| Commercial assay or kit | HDAC Fluorometric Activity Assay Kit | Cayman Chemical | 10011563 | |
| Commercial assay or kit | HDAC2 Fluorogenic Assay Kit | BPS Bioscience | 50062 | |
| Commercial assay or kit | HDAC3 Fluorogenic Assay Kit | BPS Bioscience | 50073 | |
| Commercial assay or kit | HDAC8 Fluorogenic Assay Kit | BPS Bioscience | 50068 | |
| Commercial assay or kit | HDAC4 Fluorogenic Assay Kit | BPS Bioscience | 50064 | |
| Commercial assay or kit | HDAC6 Fluorogenic Assay Kit | BPS Bioscience | 50076 | |
| Commercial assay or kit | PolyATtract mRNA Isolation System | Promega | Z5310 | |
| Commercial assay or kit | Superscript III | Invitrogen | 18080093 | |
| Commercial assay or kit | SYBR Green Master Mix | Bio-Rad | 1725270 | |
| Commercial assay or kit | TRIzol Reagent | Invitrogen | 15596026 | |
| Commercial assay or kit | TruSeq RNA Library Preparation Kit v2 | Illumina | RS-122-2001 | |
| Commercial assay or kit | Protease inhibitor cocktail | Roche | 11697498001 | |
| Commercial assay or kit | Clarity ECL Western Blotting Substrate | Bio-Rad | 1705060 | |
| Chemical compound, drug | Diacetyl | Sigma-Aldrich | B85307 | |
| Chemical compound, drug | Sodium butyrate | Sigma-Aldrich | B5887 | |
| Chemical compound, drug | Valproic acid | Sigma-Aldrich | P4543 | |
| Chemical compound, drug | Methyl pyruvate | Alfa Aesar | A13966 | |
| Chemical compound, drug | Allyl butyrate | Aldrich Chemistry | 246522-100ml | |
| Chemical compound, drug | 2,3-Butanediol | Acros Organics | 107642500 | |
| Chemical compound, drug | 2,3-Hexanedione | Alfa Aesar | L04669 | |
| Chemical compound, drug | 2,3-Heptanedione | Alfa Aesar | A19136 | |
| Chemical compound, drug | 2,3-Pentanedione | Aldrich Chemistry | 241962-25G | |
| Chemical compound, drug | 1-Acetoxyacetone | Alfa Aesar | H31346 | |
| Chemical compound, drug | Propyl formate | Sigma-Aldrich | W294306-Sample-K | |

### *Drosophila* stocks and manipulations

Fly stocks were maintained on conventional fly food under a 12 hr light:12 hr dark cycle at 18°C or 25°C. The fly strain of *w1118* backcrossed five times to *Canton-S* (*wCS*) was used in all the *Drosophila* transcriptome experiments. P{GMR-HTT.Q120}2.4 (Bloomington # 8533) were used for neurodegeneration experiments. For imaging *Gr63a-Gal4; UAS-mcd8:GFP, Gr21a-Gal4; UAS-mcd8:GFP* stocks were obtained as a kind gift from the Carlson lab, Yale.

## Odor exposure experiments for imaging

Newly eclosed male wCS flies were sorted in groups of 30 into vials with standard cornmeal medium. Vials were closed with two overlapping 3″ × 3″ polypropylene mesh squares and tied with cotton twine. Vials were used within 6 hr. Four vials were placed in a 1000 ml Nalgene straight-sided jar with a 10 ml glass beaker with 10 ml of diacetyl $10^{-2}$ in paraffin oil. Jars were closed for exposure periods ranging from 2 to 6 d. In recovery experiments, vials were removed from the jars and were placed in a 25°C incubator for the remainder of the recovery period. Flies were anesthetized on ice, sacrificed in ethanol, and then immediately put into 1× phosphate-buffered saline with 0.3% Triton-X (PTX). Brains or antenna were put in 4% paraformaldehyde in PTX and incubated for 30 min while rotating at 25°C. Samples were washed five times in PTX and blocked in 5% natural goat serum in PTX for 1 hr while rotating at 25°C. Samples were then incubated in primary antibody with 5% goat serum in PTX, mouse-nc82 (1:20) (Development Studies Hybridoma Bank, University of Iowa) and rabbit-antiGFP (1:150) (Invitrogen) in PTX for 48 hr in 4°C while rotating. After washing five times in PTX, samples were incubated in secondary antibody with 5% goat serum in PTX, rabbit-anti-Alexa488 (1:400) (Invitrogen), and mouse-anti-Alexa568 (1:400) (Invitrogen) for 48 hr in 4°C. Samples are washed five times in PTX and stored in 70% glycerol in PTX. Images were taken using a Ziess 510 laser scanning confocal microscope, and image analysis was done using ImageJ software.

## QRT-PCR

*Drosophila* heads (with antenna) were collected on liquid nitrogen and stored at –80°C until processing. A total of 300 heads were collected for each sample and ground on liquid nitrogen. mRNA was extracted using a PolyATract kit (Promega) and was reverse transcribed using Superscript III (Invitrogen) according to the manufacturer's protocol. The quality of the cDNA was assessed on a 1.5% agarose gel. cDNA was added to individual reactions of SYBR green master mix (Bio-Rad) and run on a Bio-Rad-MyQ thermocycler. The program began with a single cycle for 3 min at 95°C, followed by 40 cycles of 15 s at 95°C, 30 s at 58°C and 30 s at 72°C. Afterward, the PCR products were heated to 95°C for 1 min and cooled to 55°C for 1 min to measure the dissociation curves. The efficiency of each primer set was first validated by constructing a standard curve and three 10× serial dilutions of first-strand cDNA. For each serial dilution, the CT value was plotted against the log (template dilution) and the slope and $r^2$ value of each regression line calculated. Expression of each *Or* or *Gr* gene was assessed in triplicate. Dissociation curves were used to assess the purity of the PCR reactions. *Or* and *Gr* transcript levels were normalized by using the transcript levels of *ribosomal protein 49*. Expression levels were calculated using the Pfaffl equation.

In general, primers were designed by using coding sequences close to the 3′ end of the gene, and where possible, primers spanned an intron.

## Odor exposure protocol for transcriptome analysis

Flies were exposed to diacetyl (B85307, Sigma-Aldrich, St. Louis, MO) by placing them in vials in a cylindrical closed container (112 mm diameter × 151 mm height) along with an odor-containing glass vial. The odorant was dissolved in 5 ml paraffin oil at 1% dilution. For a given exposure protocol, two groups of flies were prepared: those exposed to 1% diacetyl headspace and those exposed to paraffin oil headspace alone (control flies). Adult male flies aged 1 d were transferred to fly vials containing fresh medium and put into the container with the odor vial. At the end of the fifth day of exposure, flies were collected, and their antennae were dissected for RNA extraction. All treatments and experiments were performed at room temperature. For the recovery experiment, flies were transferred to a container with a glass vial of paraffin oil after 5 d of diacetyl exposure. At the end of the fifth day of recovery, flies were collected, and their antennae were dissected for total RNA extraction. The second and third antennal segments from 40 to 60 male flies after treatment were carefully hand-dissected

from the head and collected in 1.5 ml microfuge tubes kept cold in liquid nitrogen. Antennae were mechanically crushed with disposable RNAse-free plastic pestles, and total RNA was isolated using a Trizol-based protocol. cDNA libraries were prepared from total RNA using the Illumina TruSeq RNA Sample Preparation Kit (v2) and 50 bps single- and paired-end sequencing was done using the HighSeq2000. Two biological replicates were sequenced for each condition, with an average of 27 million reads/ replicate, and with an average of 84% mapped.

Two-month-old C57BL/6 male mice (2–3 for each condition in a single cage) were continually exposed to air flowing over headspace of paraffin oil (solvent control), 0.1%, or 1% diacetyl over a period of 5 d, then euthanized for collecting the lung and brain tissues and processing for mRNA isolation. All protocols for animal use and euthanasia were approved by the Institutional Animal Care and Use Committee (#20150028) and were in accordance with the provisions established by the Animal Welfare Act and the Public Health Services (PHS) Policy. In the transcriptome analysis, two replicates were performed for each condition, with an average of 123,687,411 reads/replicate, with an average of 88% mapped. Multiplexed libraries were made from total RNA input using the Illumina TruSeq RNA sample preparation kit (v2) and 50 bps single-end sequencing was done using the NextSeq500.

## Calculation ppm in air based on weight loss of odorant over time

The odor solution in PO in each setup (*Drosophila* and mouse) was weighed using a sensitive balance. PO is nonvolatile and therefore weight loss is attributable to diacetyl volatilization. Volatile Headspace exposure (ppm) is calculated through odor cartridge weight change from pure odor compound volatizing out (*Inamdar et al., 2012*).

$$ppm = (10^{\wedge}6w/MW)/(v/V_m)$$

where ppm is the concentration of tested VOC in parts per million (v/v), $w$ is the weight of the tested chemical in grams, $MW$ is the molecular weight of the tested VOC (g/mole), $v$ is the total volume of the respective exposure chambers, and $V_m$ is the molecular volume under the tested condition.

$$V_m = 24.45(760/P)((t + 273.15)/298.15)$$

where 24.45 is the gram molecular volume, under standard lab conditions of 760 mmHg, 25°C; $P$ is the ambient pressure, in mmHg; and $t$ is the ambient temperature, in°C. Odor cartridges, consisting of varying odor volume/volume dilutions, were weighed periodically under standard ambient temperature and pressure. Under these conditions, the pure volatile odor compounds volatize out of the odor cartridge and the evaporation rates were used for our exposure calculations. Based on ideal gas law and experimental conditions, ppm is calculated based on volatized odor compound and experimental chamber volume based on airflow.

## HDAC inhibitor treatment protocol for transcriptome analysis

Sodium butyrate (B5887, Sigma-Aldrich) or valproic acid (P4543, Sigma-Aldrich) was dissolved in normal fly food medium at the final concentration of 10 mM. Three groups of flies were prepared: those treated with one of the HDAC inhibitors and those without HDAC inhibitor treatment (control flies). Adult flies aged 1 d were transferred to fly vials containing medium with or without a HDAC inhibitor. At the end of the fifth day of treatment, flies were collected, and their antennae were dissected for RNA extraction. All treatments and experiments were performed at room temperature. Two biological replicates with 60 flies/replicate were performed for each condition, with an average of 23 million reads/replicate, and with an average of 92% mapped. Multiplexed libraries were made from total RNA input using the Illumina TruSeq RNA sample preparation kit (v2) and 50 bps paired-end sequencing was done using the HighSeq2000.

## Bioinformatic analysis of RNA-seq experiments

Reads aligned to the latest release of each of the genomes used (dm6 for the *Drosophila* genome, GRCm38 for the *Mus musculus* genome) and quantified with kallisto (version 0.43.1) (*Bray et al., 2016*) Only libraries for which we obtained >75% alignment were used for downstream analysis. Transcript counts were summarized to gene-level using tximport package (version 1.4.0) (*Soneson et al., 2015*).

For any instances of detected batch effects, we removed unwanted variation using RuvR in the RuvSeq package (version 1.10.0) (*Risso et al., 2014*). DEG analysis was performed with the edgeR package (version 3.18.1) (*Robinson et al., 2010*) using low count filtering (cpm > 0.5) and TMM normalization. Protein classification analysis was performed with PANTHER (version 13.1) (*Mi et al., 2017*). All significance analyses of gene overlap were done using the GeneOverlap package in R package (version 1.14.0). GO enrichment analysis was performed using clusterProfiler (version 3.6.0) (*Yu et al., 2012*).

## HDAC activity assays

HDAC activity of class I HDACs (HDAC1, 2, 3, and 8) was measured with the fluorometric HDAC Activity Assay kit: HDAC1 (10011563, Cayman Chemical, Ann Arbor, MI), HDAC2, HDAC 3, and HDAC 8 (50062, 50073 and 50068, BPS Bioscience, San Diego, CA), according to the manufacturer's instructions. HDAC activity of class II HDACs (HDAC4 and HDAC6) were measured with the fluorometric HDAC Activity Assay kit: HDAC4, HDAC 4 (50064 and 50076, BPS Bioscience), according to the manufacturer's instructions. HDAC activity in these 96-well format kits was directly correlated with fluorescence produced by the enzyme and substrate interaction. Each test compound was tested in duplicates, minimum. Due to interference, concentrations of >30 mM were not used in the IC50 calculation. To account for any potential absorbance or fluorescence contributed by the test chemicals interacting with the assay substrate, all baseline background measurements of test compound + substrate control wells (everything, but no HDAC enzyme) were performed and subtracted against the respective test compound wells. It is fair to note that no significant changes in background fluorescence were seen with tested compounds in comparison to control background fluorescence. HDAC activity was calculated based on averaged maximum fluorescence from negative control wells and the averaged fluorescence in test compound wells.

$$\frac{Fluorescence_{Negative\ control} - Fluorescence_{HDACi}}{Fluorescence_{Negative\ control}} x100\% = Remaining\ HDAC\ activity\ \%$$

HDAC % inhibition was directly calculated from obtained HDAC activity %.

## Cell culture and treatment

Human embryonic kidney 293 (HEK293) cells were grown in 100 mm cell culture dishes with Dulbecco's modified Eagle's medium (DMEM) (10-013, Corning, Manassas, VA), supplemented with 10% fetal bovine serum (FBS) (26140-079, Gibco, Carlsbad, CA) at 37°C with 5% $CO_2$. Cells that were ~80% confluent were treated with freshly prepared medium supplemented with diacetyl at the concentrations indicated. The cells for mock controls were handled in the same manner without adding diacetyl to the medium. To prevent diffusion of diacetyl odor from the treatment dishes to the ones of mock control, the cell culture dishes in different conditions were cultured in separate $CO_2$ incubators.

Cell proliferation assays were performed to assess the effects of compounds against cancer cell lines. Cells were maintained in a 37°C incubator with 5% $CO_2$ throughout the whole experiment. Compound stocks are dissolved in 10% DMSO and final concentrations made fresh in complete media as needed. Then, 10% DMSO solvent is used as control in complete media. Cells were seeded onto 12-well plates and allowed 12–24 hr until treatment to allow cells to adhere to wells. This is followed by minimum 5 d treatment of compounds, changing the media with treatment every 48 hr. Cell lines A549 and SK-MEL-5 were seeded at 8000 cells/well, which was determined to provide 80–100% confluency after 6 d growth. SH-SY5Y was seeded at 8000 cells/well and 16,000 cells/well due to their known slower division rate and given at least 10 d for sufficient growth. After allotted growth times, cell counts were performed using the Countess automated cell counter. All assays had a minimum of three replicates for all conditions and analyzed through GraphPad Prism one-way ANOVA for significance. Cell lines were from ATCC: HEK-293T, A549, SH-SY5Y, and SK-MEL-5. All four cell lines were evaluated to be negative for Mycoplasma using the following tests: A549: Hoechst DNA stain (indirect) method – negative, Agar culture (direct) method – negative, and PCR-based assay – negative. The HEK-293T, SH-SY5Y, and SK-MEL-5 were determined to be negative based on PCR-based assay.

## Odor exposure protocol for Huntington's disease model flies

Flies were exposed to diacetyl in a cylindrical container (112 mm diameter × 151 mm height). Each container was tightly closed but had two holes, one of which connected to an air suction port, and

the other to a vial containing either of 5 ml paraffin oil or 5 ml 1% diacetyl in paraffin oil. A gentle suction was applied to pull the headspace from the odor or paraffin oil vials into the cylindrical structure. pGMR-HTTQ120 flies were maintained at 18°C. Adult flies aged 1 d were transferred to fly vials containing fresh medium and put into the odor-filled container at room temperature. Paraffin oil and 1% diacetyl solution were prepared and replaced every day. At the end of the fifth day of exposure, half of the flies were collected and subjected to pseudopupil analysis as before (*Steffan et al., 2001*). The remaining flies were transferred to fresh medium and exposed to the odors for an additional 5 d. All treatments and experiments were performed at room temperature. The number of rhabdomeres in each ommatidium was counted using the pseudopupil analysis, and statistical significance was determined by unpaired $t$-test (two-tailed) against PO at each time point.

## Preparation of nuclear extracts from HEK293 cells

Nuclear extracts of HEK293 cells were prepared according to a protocol described previously (*Andrews and Faller, 1991*), with minor modifications. In brief, HEK293 cells were washed twice with cold phosphate-buffered saline (PBS) and lysed with hypotonic buffer (10 mM HEPES-KOH [pH 7.9], 1.5 mM $MgCl_2$, 10 mM KCl, protease inhibitor cocktail [04693159001, Roche, Indianapolis, IN], 1 mM DTT, 1 mM TSA). Following a brief centrifugation, the pellet was resuspended in hypertonic buffer (20 mM HEPES-KOH [pH 7.9], 25% glycerol, 420 mM NaCl 1.5 mM $MgCl_2$, 0.2 mM EDTA, protease inhibitor cocktail, 1 mM DTT, 1 mM TSA). The supernatant was recovered as nuclear extract.

## Western blot analysis

Proteins in the nuclear extracts (60 µg protein) were separated by SDS-PAGE gels (456-1043, Bio-Rad, Hercules, CA), transferred onto PVDF membranes (162-0174, Bio-Rad), and incubated with anti-histone antibodies: acetylated H3K9 (1/2000: ab4441, abcam, Cambridge, MA), acetylated H3K14 (1/5000: 06-911, EMD Millipore, Billerica, MA), and acetylated H4K5 (1/2000: 07-327, EMD Millipore). Bound antibody was detected by horseradish peroxidase-conjugated anti-rabbit secondary antibody (1/20,000: 1705046, Bio-Rad) and developed using Clarity Western ECL Substrate (1705060, Bio-Rad). Signals were detected and captured using ImageQuant LAS 4000 mini (GE Healthcare, Pittsburgh, PA), and band intensities were quantified with ImageJ software. H3K9 acetylation intensity in individual lanes was reported relative to the normalized Mock treatment and calculated using this formula: relative H3K9ace intensity for each timepoint = (H3K9ace/PCNA)/averaged (Mock H3K9ace/Mock PCNA). Statistical significance was determined by unpaired $t$-test (two-tailed) against mock at each time point.

## Materials availability

All newly created reagents in this publication such as primers are available upon request.

## Acknowledgements

We would like to thank Drs. Maurizio Pellecchia, Parima Udompholkul, Francis Sladek, and Xuan Liu for cell lines. We would like to thank Dr. Larry Marsh for advice about the fly eye assay and HDAC inhibitors, Dr. Joel Kowalewski for help with critical comments, Dr. Thomas Girke for guidance in bioinformatics analysis, Drs. Anupama Dahanukar, Naoki Yamanaka, Chih-cheng Tsai, Karine LeRoch. and Adler R Dillman for sharing equipment and reagents. This work was partially supported by funds provided by the National Institutes of Health, NIAID (R01AI153195 to MGN and R01AI087785 to SHY, RNF, CAS, SP, STC, CP and AR), and HATCH funds from the Agricultural Experimentation Station UC Riverside to AR.

## Additional information

### Competing interests

Rogelio Nunez-Flores: Inventor on a pending patent application PCT/US2023/016808. Anandasankar Ray: Founder and President of Remote Epigenetics Inc and Sensorygen Inc, Remote Epigenetics Inc is involved in developing agricultural applications with volatiles. Inventor on a pending patent

application related to some data here in PCT/US2023/016808. The other authors declare that no competing interests exist.

## Funding

| Funder | Grant reference number | Author |
| --- | --- | --- |
| National Institute of Allergy and Infectious Diseases | R01AI087785 | Sachiko Haga-Yamanaka<br>Rogelio Nunez-Flores<br>Christi A Scott<br>Sarah Perry<br>Stephanie Turner Chen<br>Crystal Pontrello<br>Anandasankar Ray |
| National Institute of Allergy and Infectious Diseases | R01AI153195 | Meera G Nair |
| University of California, Riverside | | Anandasankar Ray |

The funders had no role in study design, data collection and interpretation, or the decision to submit the work for publication.

## Author contributions

Sachiko Haga-Yamanaka, Data curation, Formal analysis, Investigation, Methodology, Writing – review and editing; Rogelio Nunez-Flores, Data curation, Formal analysis, Investigation, Methodology; Christi A Scott, Data curation, Formal analysis, Investigation; Sarah Perry, Data curation, Formal analysis, Methodology; Stephanie Turner Chen, Data curation, Formal analysis; Crystal Pontrello, Formal analysis, Investigation; Meera G Nair, Supervision, Writing – review and editing; Anandasankar Ray, Conceptualization, Supervision, Funding acquisition, Methodology, Writing – original draft, Project administration, Writing – review and editing

## Author ORCIDs

Sachiko Haga-Yamanaka (ID) http://orcid.org/0000-0002-4101-9889
Meera G Nair (ID) http://orcid.org/0000-0002-1807-5161
Anandasankar Ray (ID) https://orcid.org/0000-0003-4133-2581

## Ethics

All protocols for animal use and euthanasia were approved by the Institutional Animal Care and Use Committee (#20150028) and were in accordance with the provisions established by the Animal Welfare Act and the Public Health Services (PHS) Policy.

Joint Public Review: https://doi.org/10.7554/eLife.86823.3.sa1
Author Response https://doi.org/10.7554/eLife.86823.3.sa2

# Additional files

## Supplementary files
• MDAR checklist

## Data availability

Sequencing data have been deposited in GEO under accession code GSE116502.

The following dataset was generated:

| Author(s) | Year | Dataset title | Dataset URL | Database and Identifier |
| --- | --- | --- | --- | --- |
| Haga-Yamanaka S, Scott CA, Perry S, Pontrello C, Goh M, Ray A | 2020 | A conserved odor detection pathway via modulation of chromatin and cellular gene expression | https://www.ncbi.nlm.nih.gov/geo/query/acc.cgi?acc=GSE116502 | NCBI Gene Expression Omnibus, GSE116502 |

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
