## [Editor Report · eLife assessment]

This interesting and **important** work shows that diacetyl, a volatile organic compound released by yeast in fermenting fruit, can act as a histone deacetylase (HDAC) inhibitor and trigger wide changes in gene expression, together with suppression neurotoxicity in a *Drosophila* model of Huntington's disease. While the effects on gene expression changes and degenerative phenotypes are **convincingly** shown, further studies are required to determine whether and how olfactory sensory neurons and odorant receptors mediate the effects of diacetyl described by the authors.

---

## [Referee Report · Joint Public Review]

Yamanaka et al.'s research investigates into the impact of volatile organic compounds (VOCs), particularly diacetyl, on gene expression changes. By inhibiting histone acetylase (HDACs) enzymes, the authors were able to observe changes in the transcriptome of various models, including cell lines, flies, and mice. The study reveals that HDAC inhibitors not only reduce cancer cell proliferation but also provide relief from neurodegeneration in fly Huntington's disease models. The revised manuscript addresses the key queries raised in the initial reviews.

---

## [Author Response]

The following is the authors’ response to the original reviews.

**Public Reviews:**

**Reviewer #1 (Public Review):**
Yamanaka et al.'s research investigates into the impact of volatile organic compounds (VOCs), particularly diacetyl, on gene expression changes. By inhibiting histone acetylase (HDACs) enzymes, the authors were able to observe changes in the transcriptome of various models, including cell lines, flies, and mice. The study reveals that HDAC inhibitors not only reduce cancer cell proliferation but also provide relief from neurodegeneration in fly Huntington's disease models. Although the findings are intriguing, the research falls short in providing a thorough analysis of the underlying mechanisms.HDAC inhibitors have been previously shown to induce gene expression changes as well as control cell division and demonstrated to work on disease models. The authors demonstrate diacetyl as a prominent HDAC inhibitor. Though the demonstration of diacetyl is novel, several similar molecules have been used before.

In this manuscript we are not trying to understand the mechanisms by which HDAC inhibitors affect Huntington’s disease or cancer, since these have either been studied in detail before and are outside the scope of this manuscript. Our focus is to demonstrate that volatile odorants commonly found in the environment can inhibit HDACs, alter gene expression, and have downstream physiological effects. To the best of our knowledge this unusual effect of odorants has not been systematically described before.

**Reviewer #2 (Public Review):**
Sachiko et al. study presents strong evidence that implicates environmental volatile odorants, particularly diacetyl, in an alternate role as an inhibitors HDAC proteins and gene expression. HDACs are histone deacetylases that generally have repressive role in gene expression. In this paper the authors test the hypothesis that diacetyl, which is a compound emitted by rotting food sources, can diffuse through blood-brain-barrier and cell membranes to directly modulate HDAC activity to alter gene expression in a neural activity independent manner. This work is significant because the authors also link modulation of HDAC activity by diacetyl exposure to transcriptional and cellular responses to present it as a potential therapeutic agent for neurological diseases, such as inhibition of neuroblastoma and neurodegeneration.The authors first demonstrate that exposure to diacetyl, and some other odorants, inhibits deacetylation activity of specific HDAC proteins using in vitro assays, and increases acetylation of specific histones in cultured cells. Consistent with a role for diacetyl in HDAC inhibition, the authors find dose dependent alterations in gene expression in different fly and mice tissues in response to diacetyl exposure. In flies they first identify a decrease in the expression of chemosensory receptors in olfactory neurons after exposure to diacetyl. Subsequently, they also observe large gene expression changes in the lungs, brain, and airways in mice. In flies, some of the gene expression changes in response to diacetyl are partially reversable and show an overlap with genes that alter expression in response to treatment with other HDAC inhibitors. Given the use of HDAC inhibitors as chemotherapy agents and treatment methods for cancers and neurodegenerative diseases, the authors hypothesize that diacetyl as an HDAC inhibitor can also serve similar functions. Indeed, they find that exposure of mice to diacetyl leads to a decrease in the brain expression of many genes normally upregulated in neuroblastomas, and selectively inhibited proliferation of cell lines which are driven from neuroblastomas. To test the potential for diacetyl in treatment of neurodegenerative diseases, the authors use the fly Huntington's disease model, utilizing the overexpression of Huntingtin protein with expanded poly-Q repeats in the photoreceptor rhabdomeres which leads to their degeneration. Exposing these flies to diacetyl significantly decreases the loss of rhabdomeres, suggesting a potential for diacetyl as a therapeutic agent for neurodegeneration.The findings are very intriguing and highlight environmental chemicals as potent agents which can alter gene expression independent of their action through chemosensory receptors.

We thank the reviewer for the encouraging comments.

**Recommendations for the authors:**

**Reviewer #1 (Recommendations For The Authors):**
1. The results section for figure 1 seems poorly written with errors in figure citations. Please rewrite this section.

We thank the reviewer for pointing it out and have now rewritten the results section as well as made concomitant changes in the introduction to address this comment.

1. Discussion could be more focused and could speculate mechanistic details of HDAC inhibitors in rescue of neurodegeneration.

We have added in information about the mechanistic role of the HDAC inhibition in rescue of neurodegeneration. “Exposure to diacetyl volatiles in the fly model of Huntington’s disease reduces cell degeneration, as has been previously observed with orally administered HDAC inhibitors like sodium butyrate and SAHA in this genetic model (27). Previous studies indicate that the inhibition of HDACs counter the acetyltransferase inhibitory activity of the polyglutaminedomain of the human Htt protein which binds to p300, P/CAF and CBP (27).”

A few minor comments are:1. Figure 1 is not properly cited in the test (Eg: line 137- Its not relevant to Fig 1B and its to IC)

We thank the referee for pointing out our error and have now corrected it.

1. Some Abbreviations were not expanded at the first sight, which made difficult in understanding the statement Eg: Line 51- VOC, 111- Or

We have now defined abbreviations the first time they appear in the manuscript.

1. Line 98- What was the unit when you mention 0.01%?

We have added (v/v) in the text to represent the standard volume / total volume. We have also described it in the method section.

1. Line 138- there is no comparative study done with b-HB, but the authors have claimed its was comparable. If it’s from previous study, a relative comparative statement could be given.

We apologize for the confusion. We have added the IC50 values previously reported for b-hydroxy butyrate “IC50 for HDAC1: 5.3 mM and HDAC3 2.4 mM” which was shown in the reference #21.

1. In lines 146-150, more details of what are the compounds and how similar they are to diacetyl could be added

We have added representative structures and names for the chemicals tested in Figure 1C.

1. In line 160, Why specifically they increase H3K14 acetylation?

This observed increased H3K9 (not H3K14) acetylation levels is identical to what has previously reported for b-hydroxybutyrate. We have added a sentence pointing out this similarity “preferable acetylation of H3K9 was also observed in HEK193 cells with b-hydroxybutyrate (reference #21)”.

1. In line 317, How HDAC inhibitors reverse the PolyQ disorder? What is its mechanism? Can at least discuss in the discussion section.

Our assay is based on a previous publication using the *Drosophila* model (Ref #27) and evaluated the mechanisms in detail. We have now added a section in the Discussion describing the past findings. “Exposure to diacetyl volatiles in the fly model of Huntington’s disease reduces cell degeneration, as has been previously observed with orally administered HDAC inhibitors like sodium butyrate and SAHA in this genetic model (27). Previous studies indicate that the inhibition of HDACs counter the acetyltransferase inhibitory activity of the polyglutamine-domain of the Htt protein which binds to p300, P/CAF and CBP (27).”

1. In figures, 1C and 1D, proper labeling of drug molecules is missing. Check 1D- Could have included Diacetyl for comparison, Where is the uninhibited control (negative)?

We have added the name of the chemical compounds to Figure 1C and 1D. Each compound tested has a separate blank control, which forms the basis for calculation of the percentage inhibition. The negative control is therefore part of each column.

**Reviewer #2 (Recommendations For The Authors):**
As specific feedback for the authors, I have a few questions/recommendations about the main point of the paper:a. Throughout the manuscript, the authors demonstrate gene expression differences in different tissues in flies and mice in response to exposure to diacetyl using both transgenic reporter expression and RNAseq. The authors mention they were able to show that these gene expression changes are independent of neural activity, yet I am not sure which experiment specifically demonstrates this. How do the authors know that these changes in gene expression are due to diacetyl reaching the brain after passing blood brain barrier but not due to changes in gene expression with olfactory circuit activity? I acknowledge that disproving that the gene expression differences are independent of neural activity, but one question is whether inhibiting neural activity result in changes in the expression of overlapping genes in the same direction. Or for example, if one inhibits neural activity in Gr21a neurons, do they reversibly shut down expression of the receptor after a few days? Is this true for other ORs or specific to Gr21a and Gr63a?

While it is difficult to completely rule out contributions of the olfactory effects in the brain, we also report differential gene expression in the lungs of mice where we do not expect olfactory circuit activity (Fig 3D-G). The overlap in DEGs is highly statistically significant between the organs suggesting at least some commonality in mechanism (Fig 5D). We recently evaluated a *Drosophila* tissue that does not express odorant receptors or connections, the ovaries, and also found substantial evidence of diacetyl-exposed modulation of genes. While the data are intended for a different publication, we found up to 123 up and 61 downregulated DEGs (FDR cutoff <0.05 and log2 fold change cutoff of 1 and -1). These data should also be viewed together with the in vitro HDAC inhibition data and the increased histone acetylation seen in cell lines.

b. Is diacetyl detected by any chemosensory receptors in flies or mice? RNA profiles from these receptor mutants can be used to distinguish whether the gene expression changes are occurring due to neural activity or direct ability of diacetyl to alter HDAC activity. One relatively simple experiment would be to test whether differentially expressed genes in the orco mutant antennae overlap at all with antennal RNA profiles from diacetyl exposed flies.

Diacetyl can be detected by multiple chemosensory receptors in flies and mice. In flies the Gr21a+Gr63a complex expressing neurons are inhibited by diacetyl as indicated, and Or9a, Or43b, Or59b, Or67a, and Or85b are activated receptors (Hallem, Cell, 2006). It would be extremely resource and time-consuming process to create and evaluate single mutants or combinations of mutants as suggested. In response to the previous point, we noted examples of tissues without olfactory receptors or olfactory circuits showing DEGs upon diacetyl exposure.

As suggested by the referee, we compared DEGs from RNASeq data of Orco mutant antenna (N=2 replicates) generated for another project. There is very little overlap between antennal DEGs from Orco and the diacetyl (labelled chart as d4on_up and d4on_down) exposed flies. These data suggest that large-scale silencing of antennal neurons in Orco mutants do not alter expression of the same genes as altered by exposure to diacetyl.

**Author response image 1. sa2fig1:** 

c. The comparison of DEGs from individuals exposed to diacetyl versus the other two HDAC inhibitors shows some overlap. The overlap is greater for DEGs shared between the two HDAC inhibitors. Yet, there is still a substantial number of genes that are unique to diacetyl exposure. For example, if you compare SB to VA exposure, each condition has about 150-200 genes uniquely misexpressed for each condition with about 55 genes shared. However, the number of uniquely misexpressed genes is over 600 for diacetyl exposed individuals, with only 30 and 100 genes shared with either SB and VA respectively. I would have expected a higher overlap in DEGs if these compounds all inhibit similar HDACs. Do they inhibit different HDACs? Can this explain the significant number of uniquely misexpressed genes in each condition?

It is difficult to judge significance of overlap in DEG sets the genome has around 13,000 genes from evaluating numbers without statistical analysis which we noted in the text. “A pairwise analysis using the Fisher’s exact test of each gene set revealed a statistically significant overlap of diacetyl-induced genes with SB-induced genes (p=6x10-11) and with VA-induced genes (p=2x10-65) (Figure 4F).”

We have also further clarified in the text “This highly significant overlap among upregulated genes lends further support to our model that diacetyl vapors act as an HDAC inhibitor in vivo. As expected, each of the 3 treatments also modulated a substantial number of unique genes (Figure 4G,H), suggesting that differences in delivery format (oral vs vapor delivery), molecular structure and inhibition profile across the repertoire of HDACs may contribute to differences in gene regulation.”

d. The authors show changes in RNA profiles in response to diacetyl exposure in different tissues and suggest that these are due to changes in histone acetylation without direct comparison of genes that show up or down regulation with acetylation patterns. They do show in the beginning that diacetyl inhibits HDAC function in vitro and in cell culture. Yet it is critical that they also show a general increase in acetylation levels within tissues profiled for RNA. Additional experiments profiling chromatin and histone acetylation patterns in the tissues where RNA is profiled from would strengthen the argument of the paper.

We agree with the referee’s suggestion and appreciate it. However, given the heterogeneity of the cell types and therefore histone marks in chromatin within the tissues that we analyzed, we estimate that it will require substantial effort to purify or enrich specific cell populations before performing Chip-Seq. Such studies will examine correlations between up- and down-regulated genes and histone acetylation pattens in cells in the future studies. This effort will require significant resources and time which we feel are outside the scope of this manuscript.

e. The rhabdomere experiments might benefit from a negative control. Can the authors expose the flies to another volatile and show neurodegeneration is not affected?

We exposed the negative control group to headspace odorants of paraffin oil which is a mixture of hydrocarbons.

f. The same is true for the initial HDAC activity profiles from Figure 1. Can the authors show an HDAC activity that is not affected by diacetyl exposure?

We exposed the negative control group to headspace odorants of paraffin oil which is a mixture of hydrocarbons. Diacetyl shows very little inhibition (Average inhibition = 7.69%; N=2) in purified human HDAC4 when tested at the 15mM concentration.

g. One point that might require some explanation in the discussion is why diacetyl exposure only increases acetylation of certain histones but not others in Figure 2, especially given that many HDACs are inhibited by diacetyl in Figure 1.

Please see response to comment #6, Reviewer 1.

h. Figure S1C is missing descriptions of what different histogram colors signify.

We apologize for the oversight and have now indicated it in the Figure legend.